environmental science/environmental chemistry/chemical engineering

corn stalk, SET-LRP, adsorbent, heavy metal ions

**Author for correspondence:**
Yazhen Wang
e-mail: yzwang2957@163.com

This article has been edited by the Royal Society of Chemistry, including the commissioning, peer review process and editorial aspects up to the point of acceptance.

# Corn stalk as starting material to prepare a novel adsorbent via SET-LRP and its adsorption performance for Pb(II) and Cu(II)

Yazhen Wang[1,2,3], Shuang Li[1,3], Liqun Ma[1], Shaobo Dong[1,3] and Li Liu[3]

[1]College of Materials Science and Engineering, Qiqihar University, Qiqihar 161006, Heilongjiang, People's Republic of China
[2]College of Chemistry, Chemical Engineering and Resource Utilization, Northeast Forestry University, Harbin 150040, Heilongjiang, People's Republic of China
[3]Heilongjiang Province Key Laboratory of Polymeric Composition Material, Qiqihar 161006, Heilongjiang, People's Republic of China

YW, 0000-0002-5470-0200; SL, 0000-0001-7275-0730

Corn stalk was used as the initial material to prepare a corn stalk matrix-g-polyacrylonitrile-based adsorbent. At first, the corn stalk was treated with potassium hydroxide and nitric acid to obtain the corn stalk-based cellulose (CS), and then the CS was modified by 2-bromoisobutyrylbromide (2-BiBBr) to prepare a macroinitiator. After that, polyacrylonitrile (PAN) was grafted onto the macroinitiator by single-electron transfer living radical polymerization (SET-LRP). A novel adsorbent AO CS-g-PAN was, therefore, obtained by introducing amidoxime groups onto the CS-g-PAN with hydroxylamine hydrochloride ($NH_2OH \cdot HCl$). FTIR, SEM and XPS were applied to characterize the structure of AO CS-g-PAN. The adsorbent was then employed to remove Pb(II) and Cu(II), and it exhibited a predominant adsorption performance on Pb(II) and Cu(II). The effect of parameters, such as temperature, adsorption time, pH and the initial concentration of metal ions on adsorption capacity, were examined in detail during its application. Results suggest that the maximum adsorption capacity of Pb(II) and Cu(II) was 231.84 mg g$^{-1}$ and 94.72 mg g$^{-1}$, and the corresponding removal efficiency was 72.03% and 63%, respectively. The pseudo-second order model was more suitable to depict the adsorption process. And the adsorption isotherm of Cu(II) accorded with the Langmuir model, while the Pb(II) conformed better to the Freundlich isotherm model.

# 1. Introduction

Recently, heavy metal ions have received considerable concern as the main source of the aquatic environmental pollution due to their high toxicity and non-biodegradable [1,2]. Heavy metal ions can seriously damage the human central nervous system and organisms [3,4]. Overexposure to lead causes stomach ache, headache, trembling, nervous irritability, severe convulsions and even death. Excessive copper ions can cause serious damage to people's liver and gall-bladder [5]. A tremendous amount of effort has been made to develop technologies for removal of heavy metal ions from wastewater, such as reverse osmosis chemical precipitation, photocatalysis technology, adsorption and membrane filtration [6–9]. Among all these technologies, adsorption is predominant in the sewage disposal process because of its great performance in energy saving and high efficiency [10]. Although adsorption technology is a preferred method for heavy metal ions removal, its widespread use is restricted due to the large cost and high regeneration capital [11]. A large number of attempts have been made to develop comparatively cheap materials to prepare low-cost adsorbents [12–14]. Biological macromolecules such as chitosan and cellulose are considered to be the ideal raw material for adsorbent because of their low-cost and comprehensive sources [15–17]. Currently, a wide variety of crop residue including wheat straw [18,19], sweet potato starch [20,21], sunflower leaves [22] and water bamboo husk [23] have been used as raw materials for the adsorbent.

Controlled/living radical polymerization includes nitroxide-mediated radical polymerization (NMRP), reversible addition-fragmentation chain transfer polymerization (RAFT) and atom transfer radical polymerization (ATRP) [24–26], which are the most commonly used methods for surface chemical modification of biopolymers such as cellulose, chitosan and starch. In recent years, there has been a growing concern about the single-electron transfer living radical polymerization (SET-LRP) because of its numerous advantages such as mild reaction, low reaction temperature, facile removal of the catalyst and oxygen insensitivity [27–29].

Research on the adsorbent which treated crop residue as a raw material has been conducted in the past; however, the report pertinent to the amidoxime-functionalized adsorbent prepared from corn stalk by SET-LRP and used for removal of Pb(II) and Cu(II) remains limited. As the main agricultural residue, corn stalk consists mainly of cellulose, hemicellulose and lignin, of which the cellulose content is 39%. Corn stalk could be observed to have a condensed composition of fibre structure under a scanning electron microscope; it has a reported fibre length of 1.32 mm, fibre width of 24.3 mm, lumen width of 24.3 mm and cell wall thickness of 6.8 mm [30]. Cellulose contains a high amount of hydroxyl groups which can chelate with heavy metal ions. However, the interaction between hydroxyl groups and heavy metal ions is weak; in order to increase its adsorption ability, some specific functional groups have to be introduced to the surface of corn stalk using hydroxyl groups [31–38]. On such a basis, this study researched the preparation of a corn stalk-g-polyacrylonitrile-based adsorbent via SET-LRP and verified its application for adsorption of Pb(II) and Cu(II) from wastewater. This study intends to make contributions to solving the water pollution caused by heavy metal ions by means of transforming agricultural residues into adsorbent, avoiding the pollution due to stalk burning, and treating aqueous solution pollution effectively.

# 2. Materials and methods

Corn (maize) stalk was collected from a farm in Qiqihar city, Hei Longjiang Province (China). Acrylonitrile (AN) was purchased from Tianjin Fuchen Chemical Reagents (Tianjin, China). N,N-dimethylformamide (DMF), methanol and trimethylamine (TEA) were supplied by Kermel Chemistry (Tianjin, China). Ethanol and nitric acid were obtained from Tianli Chemical Reagents (Tianjin, China). 4-Dimethylaminopyridine (DMAP), N,N,N′,N″,N″-pentamethyldlethylenetrlamlne (PMDETA), 2-bromoisobutyrylbromide (2-BiBBr) and Cu(0) powder were bought from Aladdin Chemistry (Shanghai, China). Fourier Transform infrared spectroscopy (FTIR) was recorded on a Perkin-Elmer Spectrum One B type to determine the groups which were introduced onto the surface of the adsorbents. Surface morphologies of samples were observed by a scanning electron microscope (SEM, S-3400). The elemental composition was analysed by a Vario EL cube elemental analyser. The definition of the total exchange capacity (TEC) was given on the basis of the content of bromine and nitrogen on the surface of samples. TEC values were calculated according to the following equations:

$$TEC = \frac{N\%}{1.4}$$

(2.1)

**Scheme 1.** Synthetic routes for the preparation of AO CS-g-PAN.

and

$$TEC = \frac{Br\%}{8} \ . \tag{2.2}$$

Thermogravimetric curves of corn stalk powder, grafted polymer and adsorbent were examined by an STA449F3 synchronous thermal analyser. The ICE-3500 atomic absorption spectrophotometer (AAS) was used to measure the concentration of heavy metal ions. The point of the zero charge ($pH_{pzc}$) of the adsorbent was measured using the solid addition method [39]. Brunauer–Emmett–Teller (BET) surface area and pore structure parameters of the adsorbent were determined from the nitrogen adsorption/desorption measurement at 77 K with an Autosorb-iQ surface analyser (Quantachrome Instruments, USA); samples were degassed at 120°C to remove the gases.

## 2.1. Preparation of adsorbent AO CS-g-PAN

The absorbent AO CS-g-PAN was prepared according to the synthetic routes showed in scheme 1, and the corresponding procedures were demonstrated in detail as follows.

### 2.1.1. Pretreatment of corn stalk

Corn stalks were washed with distilled water after removing leaves, then dried at 50°C, after which the clean corn stalks were crushed into granules. Then, 4.5 g of the corn stalk powder was soaked at first in 150 ml of 15% KOH solution for 2 h at 50°C and later filtered. Then, the filtered product was stirred in 200 ml of 1 mol l$^{-1}$ HNO$_3$ at 70°C for 1 h. After a while, the mixture was washed and dried, eventually obtaining the corn stalk-based cellulose (CS).

### 2.1.2. Preparation of corn stalk macroinitiator

A total of 2.0 g of CS was added to 30 ml of DMF, ultrasonicated for 10 min. After that, the mixture was transferred to a triple neck bottle. Then, 2.2 g of DMAP and 2.0 g of TEA were added to the triple neck bottle, which was cooled with an ice bath. And then, 2.5 ml of 2-BiBBr dissolved in 10 ml of DMF was added dropwise into the mixture. The triple neck bottle should be charged with nitrogen and stirred for 40 min in the ice water bath. Even more, the mixture was continuously stirred at room temperature for 24 h. When reaching the required time, the product was precipitated with ethanol and further washed

with methanol/distilled water three times. Finally, the product was dried at 60°C to form a constant mass and it was named macroinitiator CS-Br.

### 2.1.3. Preparation of CS-g-PAN by surface polymerization of acrylonitrile using CS-Br via SET-LRP

Polymerization process was operated by the following procedure. Cu(0) (0.032 g) and PMDETA (0.173 g) were mixed with 20 ml of DMF, the mixture was added into a two-necked flask, bubbled with $N_2$ to eliminate oxygen. Continuously, the macroinitiator CS-Br (0.6 g) and AN (8 g) were added into the flask in succession. After undergoing $N_2$–vacuum–$N_2$ cycles three times, the sealed flask was charged with $N_2$ and the mixture was stirred under the nitrogen atmosphere for 24 h at 60°C [40]. After a defined time, the mixture was filtered and dried until it reached a constant weight at 60°C, in which way CS-g-PAN was obtained.

### 2.1.4. Synthesis of adsorbent based on CS-g-PAN using $NH_2OH \cdot HCl$

In a typical procedure, a total of 2.0 g $NH_2OH \cdot HCl$ was dissolved in a 50 ml of methanol. An exact 1.2 g of CS-g-PAN and hydroxylamine solution were transferred to a two-necked, round-bottomed flask, and then stirred for 2 h, after which the pH of mixture was adjusted to 9 with sodium hydroxide, and the reaction maintained at 65°C for 24 h for reflux condensation. At last, the production was extracted in ethanol for 12 h and dried at 60°C [41–43].

## 2.2. Adsorption experiments

In order to examine the adsorption performance for Pb(II) and Cu(II) of the absorbent, batch adsorption experiments were carried out on a model BETS-M1 shaker (Kylin-Bell Lab Instruments Co., Ltd, China) with a shaking speed of 120 r.p.m. In a general procedure, 10 mg of adsorbent was added to 20 ml of $2.0 \times 10^{-3}$ mol l$^{-1}$ metal ions solution with constant shaking for 24 h at 303.15 K. The effect of pH value on adsorption was studied as pH ranged from 2 to 7. The effect of temperature on adsorption was researched at 293.15, 298.15, 303.15, 308.15, 313.15 and 318.15 K.

Considering the effect of metal ion loss caused by glass bottles adsorption on the performance of AO CS-g-PAN, the control experiment without adsorbent was carried out in the same circumstances to determine the amount of metal ions adsorbed by the glass bottles. And all the data of adsorption experiments have eliminated the error caused by the adsorption of glass bottles.

The adsorption kinetics were carried out with the initial concentration of 414.4 mg l$^{-1}$ for Pb(II) and 128 mg l$^{-1}$ for Cu(II) at 303.15 K. Adsorption isotherm was conducted with initial concentrations ranging from 20 to 200 mg l$^{-1}$ for Pb(II) and from 15 to 130 mg l$^{-1}$ for Cu(II). And then, the concentration of Pb(II) and Cu(II) was detected by AAS. The amount of the metal ion at equilibrium was estimated by using the following equation:

$$q_e = \frac{(C_0 - C)V}{W}, \tag{2.3}$$

where $q_e$ (mg g$^{-1}$) is the equilibrium adsorption amount; $C_0$ (mg l$^{-1}$) and $C$ (mg l$^{-1}$) are the initial concentration and final concentration of metal ions, respectively; $W$ (g) is the weight of the adsorbent; and $V$ (l) is the volume of solution. In the adsorption experiments, all samples were performed three times and averaged the adsorption capacity.

# 3. Results and discussion

## 3.1. FTIR analysis

As shown in figure 1, compared with the spectrum of CS in figure 1a, the spectrum of partial acylation CS shows a new band at 1740 cm$^{-1}$ in figure 1b, which corresponds to the vibration of the carbonyl in the ester group of CS-Br. It proved that the macroinitiator CS-Br has been prepared successfully. As shown in figure 1c, a new band emerged at 2242 cm$^{-1}$ after grafting, which was attributed to the vibration of cyano groups confirmed the success of graft polymerization. Compared with figure 1c, the spectra of AO CS-g-PAN was shown in figure 1d where the success of CS-g-PAN modification was verified by both the disappearance of the bands at 2244 cm$^{-1}$ attributed to cyano groups and the

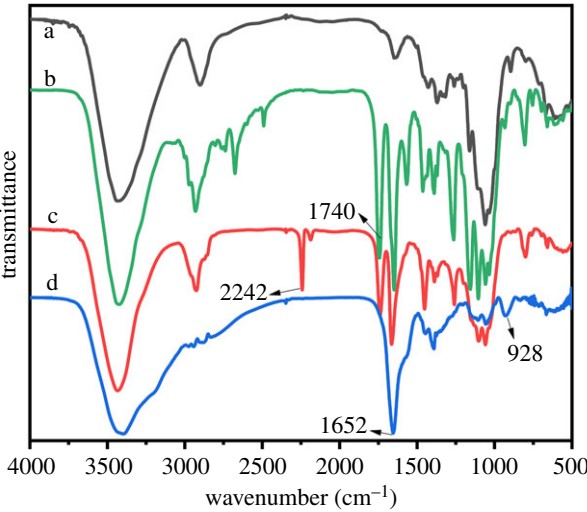

**Figure 1.** FTIR spectra of samples (a) CS, (b) CS-Br, (c) CS-g-PAN, and (d) AO CS-g-PAN.

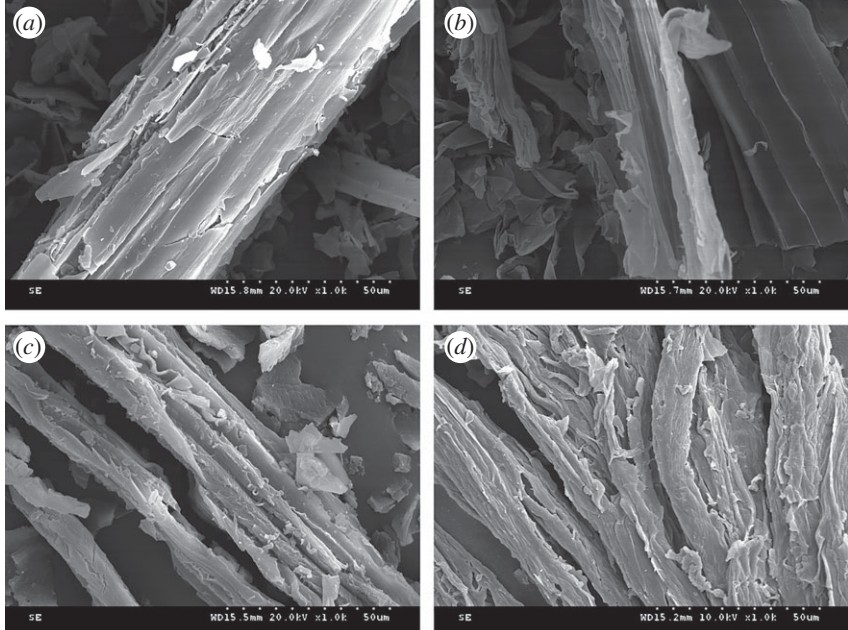

**Figure 2.** SEM micrographs of (*a*) corn stalk, (*b*) CS, (*c*) CS-g-PAN, and (*d*) AO CS-g-PAN.

emergence of two characteristic vibrations at both 1652 and 928 cm$^{-1}$, respectively, related to the stretching vibration of the C–N and N–O bonds of the amidoxime groups [44].

## 3.2. Morphology analysis

The surface morphologies of (*a*) corn stalk, (*b*) CS, (*c*) CS-g-PAN and (*d*) adsorbent AO CS-g-PAN are exhibited in figure 2. Some fibre bundles can be clearly seen in figure 2*a*. Compared with the original ones, the structure and morphology of the processed corn stalk have been obviously changed after being treated with potassium hydroxide and nitric acid; the fibre bundles were destroyed in the pretreatment reactions so that more hydroxyl groups became exposed, which is favourable to subsequent reactions, including the acylation, graft and adsorption processes. It also illustrated that the corn stalk-based cellulose was made accessible. As shown in figure 2*c*,*d*, after graft modifying, the smooth surface of corn stalk became coarse, which proved the successful preparation of CS-g-PAN. The morphology of AO CS-g-PAN observed by SEM was similar to that of CS-g-PAN, but it has a looser structure, which was advantageous to the adsorption of metal ions.

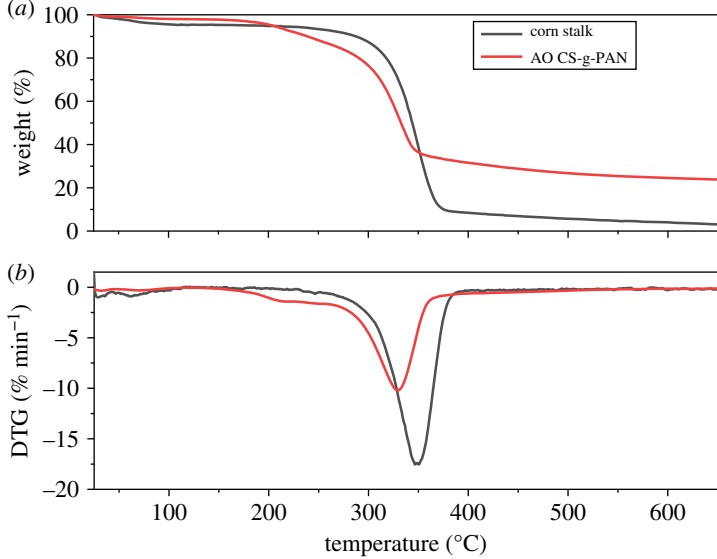

**Figure 3.** Thermogravimetric analysis (TGA) diagram (*a*) and derivative thermogravimetry (DTG) (*b*) of corn stalk and AO CS-g-PAN.

**Table 1.** Elemental analysis results of functionalized corn stalk.

| sample | O (%) | Br (%) | N (%) | TEC (mmol g$^{-1}$) |
|---|---|---|---|---|
| CS-Br | 40.28 | 0.59 | — | 0.074 |
| CS-g-PAN | 9.3 | — | 19.16 | 13.67 |
| AO CS-g-PAN | 30.56 | — | 9.35 | 6.68 |

## 3.3. Thermal degradation behaviours analysis

The thermal decomposition behaviour of corn and adsorbent were detected by a synchronous thermal analyser, and the diagram is shown in figure 3. As shown in figure 3*a*, the results revealed two platforms of the two samples in the decomposition process. The initial decomposition platform was between 50 and 200°C (56°C for stalk and 192°C for AO CS-g-PAN) caused by the loss of small molecules and monomers. By contrast, the second degradation showed a significant difference. Because of the degradation of cellulose and deacetylation, the second weight loss process emerged in the range of 200–370°C. In comparison with corn stalk, the second degradation of AO CS-g-PAN was conducted at a lower decomposition temperature, which can be ascribed to the cleavage of the C–O bond formed by the acylation reaction during the preparation of the macroinitiator. As we can see in figure 3*b*, $T_{max}$ of corn stalk and AO CS-g-PAN were 330°C and 350°C, respectively. It was important to highlight that the mass loss of corn stalk was approximately 95%, while the mass loss of AO CS-g-PAN was only 70%, the lower mass loss of AO CS-g-PAN demonstrated its higher thermal stability [45–47].

## 3.4. Elemental analysis

In order to further demonstrate the success of the modification, the Vario EL cube elemental analyser was used to determine the amount of elements on the surface of the samples, and the result of the elemental analysis is given in table 1. After graft modification, the content of N element increased from 0 to 19.16%, demonstrating that the acrylonitrile monomer had been grafted onto the surface of the macroinitiator. After the introduction of the amide oxime group, the content of O element increased from 9.3 to 30.56%, which proved that the preparation of the adsorbent was successful. The content of the organic groups on the surface of the modified corn stalks was further calculated from TEC of bromine and nitrogen. As depicted, there were 0.074 mmol g$^{-1}$ of bromine, 13.67 mmol g$^{-1}$ of cyano groups and 3.34 mmol g$^{-1}$ of amidoxime groups in CS-Br, CS-g-PAN and AO CS-g-PAN, respectively.

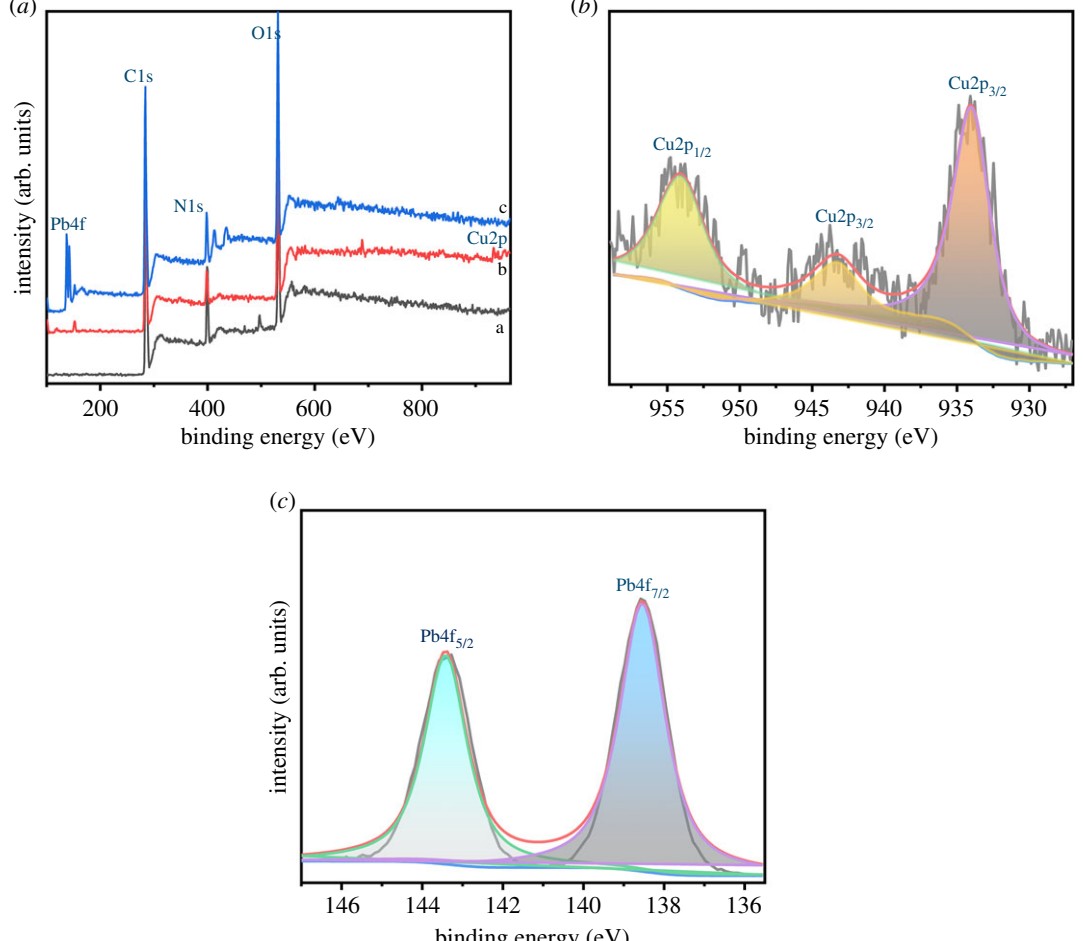

**Figure 4.** XPS spectra of (a) AO CS-g-PAN before and after adsorption, high resolution peaks of (b) Cu2p and (c) Pb4f.

**Table 2.** Pore structure parameters of CS and AO CS-g-PAN.

| sample | $S_{BET}$ (m$^2$ g$^{-1}$) | $V_{tot}$ (cm$^3$ g$^{-1}$) | pore width (nm) |
| --- | --- | --- | --- |
| CS | 2.813 | 0.084 | 15.044 |
| AO CS-g-PAN | 0.72 | 0.013 | 30.767 |

## 3.5. Surface area and pore size distribution

Surface area and porosity of CS and AO CS-g-PAN were determined by nitrogen adsorption/desorption method and the related important parameters are summarized in table 2. Based on the BET surface area of the obtained material, it does have a small surface and low porosity. The result was consistent with previous research reported by Gedam & Dongre [48], who synthesized adsorbents using chitosan and used them to remove Pb(II) from aqueous solution; the adsorbent prepared by Gedam & Dongre had a BET surface area of 0.87 m$^2$ g$^{-1}$ and pore diameter of 9.77 nm. A small surface area and almost low porosity are characteristics of natural polymer materials, such as cellulose and chitosan, but they have been used as adsorbents in the literature because of their low cost and wide availability, and the results show that their performance in metal ions removal is satisfactory [40,44].

It could be found that the graft modification actually decreased the specific surface area and the pore volume due to the blockage of internal porosity by a grafted chain of polyacrylonitrile (PAN). It was also observed that graft modification resulted in an increase in average pore diameter, indicating that the blocked pores would be micropores [49]. For the pure physical adsorption process, the adsorption capacity increases with the increasing surface area of the adsorbent [50]. Note that the untreated corn

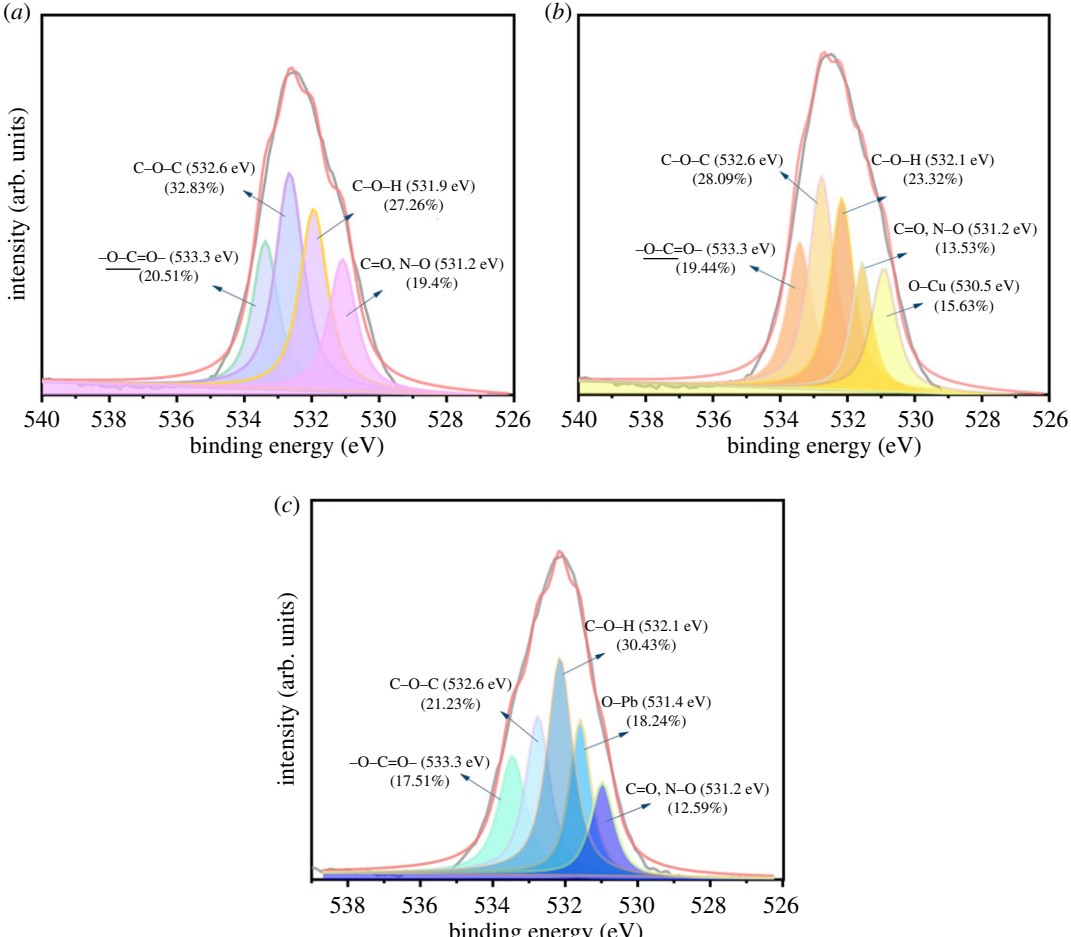

**Figure 5.** XPS spectra of O1s of (*a*) adsorbent and after loaded (*b*) Cu(II) and (*c*) Pb(II).

stalk was also investigated as an adsorbent for comparison in our initial adsorption performance test. Results suggest that the maximum adsorption capacity of AO CS-g-PAN on Pb(II) and Cu(II) were 231.84 and 94.72 mg g$^{-1}$, respectively, which is much higher than CS with the maximum adsorption capacity 64.5 and 28.32 mg g$^{-1}$. Therefore, it is speculated that the physisorption of AO CS-g-PAN for Pb(II) and Cu(II) is restricted, and the chemisorption is the predominant adsorption mechanism due to the existence of adsorption sites amidoxime groups which effective bind heavy metal ions.

## 3.6. XPS spectrum analysis

X-ray photoelectron spectroscopy (XPS) was used to further examine the adsorption mechanism. As shown in figure 4, there were main peaks C1s, O1s and N1s shown in the XPS wide-scan spectra of AO CS-g-PAN before and after loaded Cu(II) and Pb(II) [45]; furthermore, characteristic peaks emerged after the adsorption belonging to Pb4f (4f$_{7/2}$ 138.54 eV, 4f$_{5/2}$ 143.42 eV) and Cu2p (2p$_{3/2}$ 933.98 eV, 2p$_{1/2}$ 953.6 eV), respectively, which indicated that the Pb(II) and Cu(II) have been adsorbed successfully. Moreover, the O1s spectrum and N1s spectrum of the adsorbent before and after the loaded metal ions are depicted in figures 5 and 6 separately. The O1s peak of AO CS-g-PAN could be deconvoluted into four individual components: 533.3, 532.6, 531.9 and 531.2 eV assigned to O–C=O, C–O–C, C–O–H, and C=O and N–O (C=O peak and N–O peak coincided at 531.2 eV), respectively [37]. The N1s peak of AO CS-g-PAN could be deconvoluted into three individual components: 399.9, 399.5 and 398.9 eV assigned to –NH$_2$, C–N and N–O (C–N peak and N–O peak coincided at 399.5 eV), and –C=N, respectively. After adsorption, the peak of C–O–H shifted from 531.9 to 532.1 eV, and the peak of –NH$_2$ shifted from 399.9 to 400.0 eV, which revealed that C–O–H and –NH$_2$ were related to the adsorption process. In addition, new peaks at 530.5 and 531.4 eV in figure 5*b*,*c*, and 400.8 and 401.1 eV in figure 6*b*,*c* corresponding to O–Cu, O–Pb, C–N–Cu and C–N–Pb were observed [51], suggesting that Cu(II) and Pb(II) were chemically adsorbed to the adsorbent surface which N and O were likely to play the most important role in the adsorption of metals on AO CS-g-PAN from the XPS analysis [52].

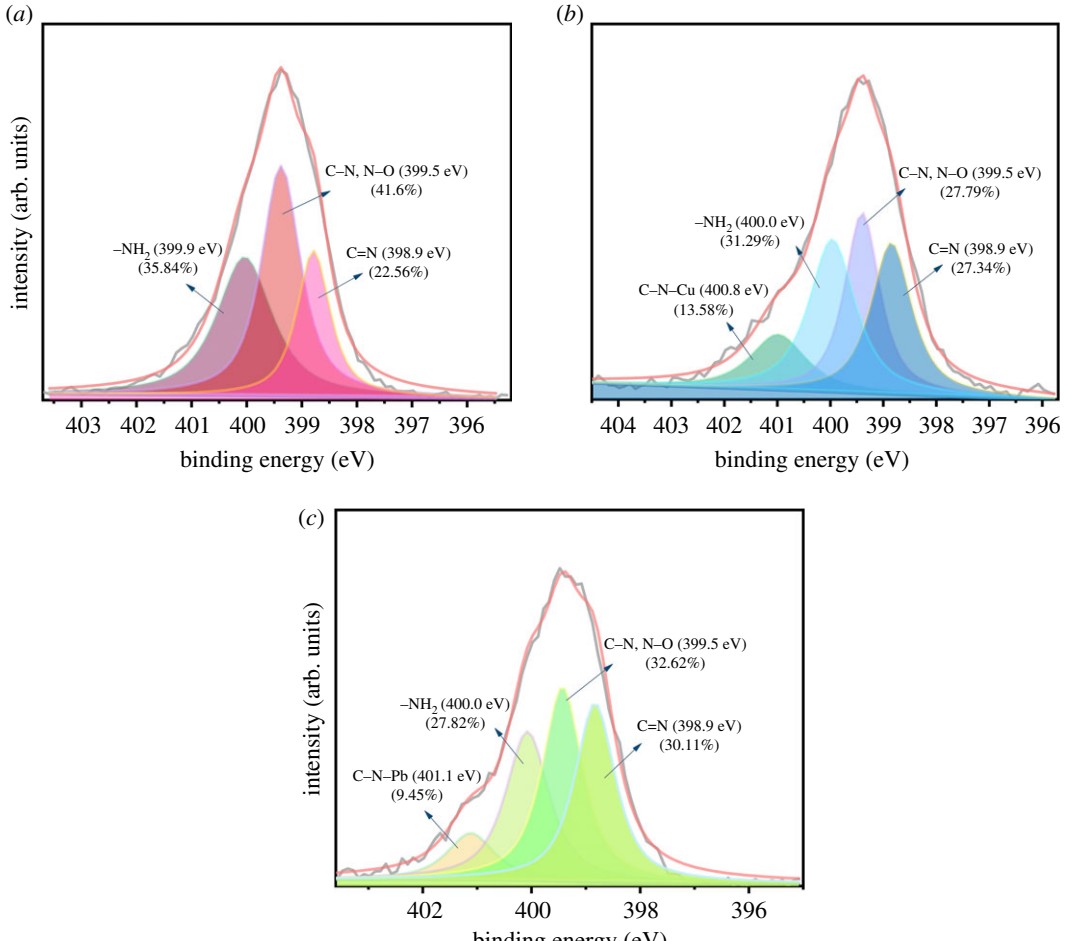

**Figure 6.** XPS spectra of N1s of (*a*) adsorbent and after loaded (*b*) Cu(II) and (*c*) Pb(II).

## 3.7. EDS spectrum analysis

The energy dispersive spectroscopy (EDS) spectra of the adsorbent before and after the adsorption of Cu(II) and Pb(II) are shown in figure 7. The adsorbent consisted of C, N and O, among which C and O were the main constituent elements. After the adsorption of Cu(II) and Pb(II), there were new peaks appearing in the energy spectra corresponding to Cu(II) and Pb(II), and the absorption intensity of Pb(II) was markedly higher than that of Cu(II). It has been confirmed that the metal ions were loaded onto the surface of the adsorbent successfully.

## 3.8. Comparison of adsorption capacity property with CS, CS-Br and CS-g-PAN

In order to reveal the adsorption performance of modified corn straw, the adsorption capacities of CS, CS-Br and CS-g-PAN on Pb(II) and Cu(II) were detected to compare with AO CS-g-PAN. Figure 8 shows the adsorption capacity and metal ions removal efficiency of CS, CS-Br, CS-g-PAN and AO CS-g-PAN. As shown in figure 8, compared with CS, the adsorption capacity of macromolecular initiator has no significant change. After the grafting of polyacrylonitrile, the adsorption capacity was improved, indicating that the cyano group might have some effect on the adsorption process. After further modification, the adsorption capacity of AO CS-g-PAN was further improved, indicating that the amidoxime group played a major role in the adsorption of Pb(II) and Cu(II). The data suggest that AO CS-g-PAN has great potential to remove heavy metal ions from wastewater.

## 3.9. Effect of pH on adsorption capacity and the characteristics of surface charge

The point of the zero charge ($pH_{pzc}$) is a very significant indicator which plays a vital role in adsorption phenomena [53]. In order to determine the $pH_{pzc}$, the pH value in which the electrical charge density of

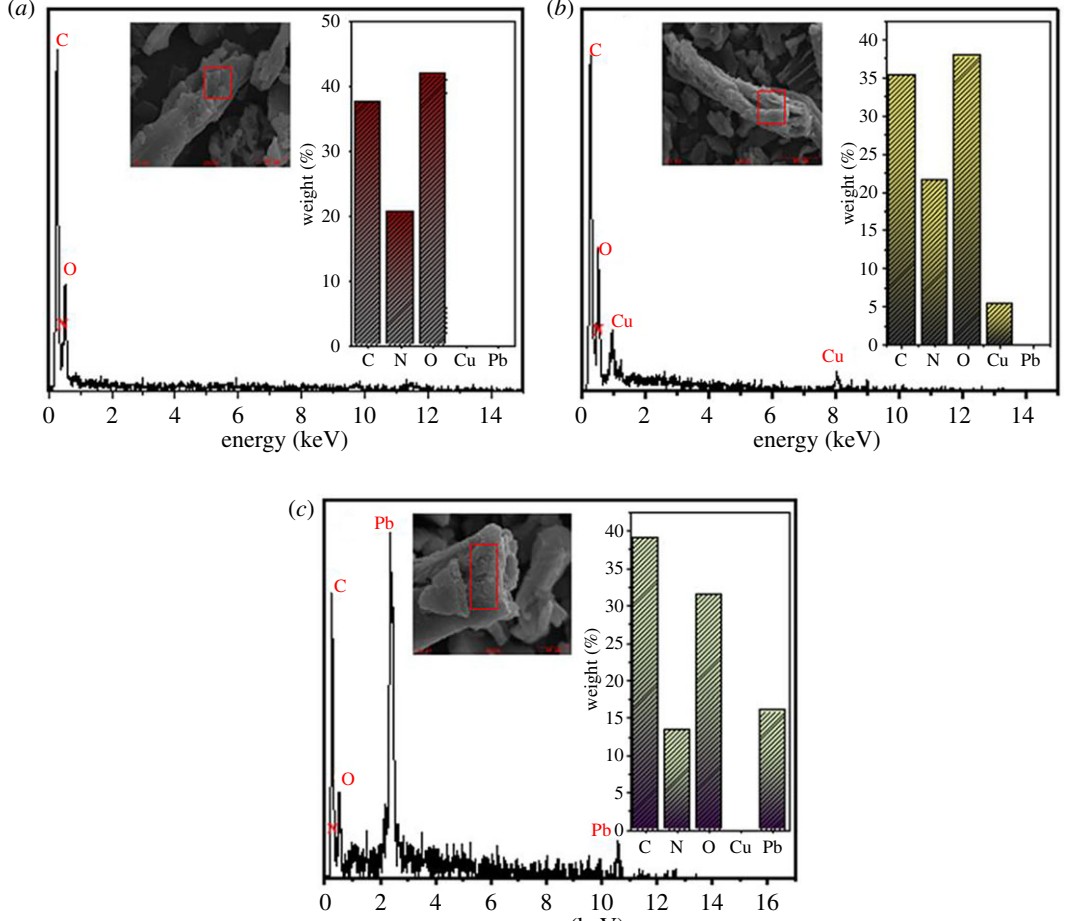

**Figure 7.** EDS spectrum of (*a*) AO CS-g-PAN and (*b*) Cu(II) and (*c*) Pb(II) loaded the adsorbent.

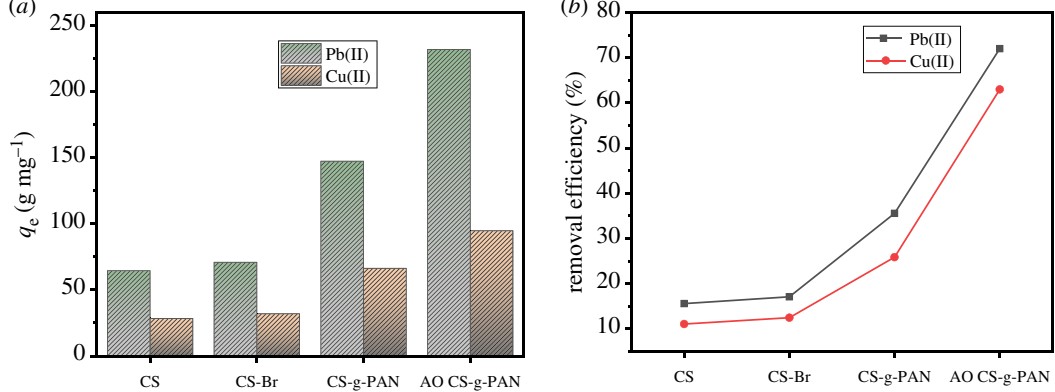

**Figure 8.** Adsorption capacity (*a*) and metal ions removal efficiency (*b*) of CS, CS-Br, CS-g-PAN and AO CS-g-PAN.

the adsorbent surface is zero, 20 mg of adsorbent was stirred with 20 ml of 0.1 mol $l^{-1}$ NaNO$_3$ solution, HNO$_3$ or NaOH were added to the mixture to adjust the pH (pH$_i$) values ranging from 1 up to 9, and the suspensions were stirred for 48 h. The final pH (pH$_f$) of suspensions was measured and the plot pH$_{final}$ and pH$_{initial}$ was obtained.

The effect of initial pH on adsorption of Cu(II) and Pb(II) was explored in the pH range from 2 to 7, and the variation tendencies of the adsorption capacity of the AO CS-g-PAN at different pH are exhibited in figure 9*a*. It illustrates that the adsorption process was pH-dependent. The pH$_{pzc}$ was defined as the pH value in which the final pH value equals the initial pH value. As shown in figure 10, the pH$_{pzc}$ was obtained at the intersection of the fitting line and straight line. Therefore, the pH$_{pzc}$ is 4.13 for adsorbent

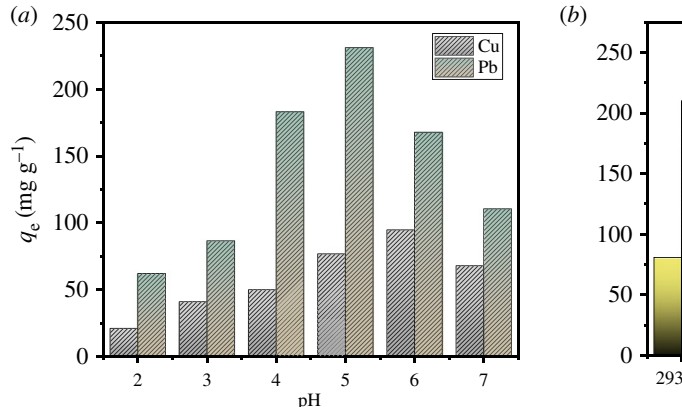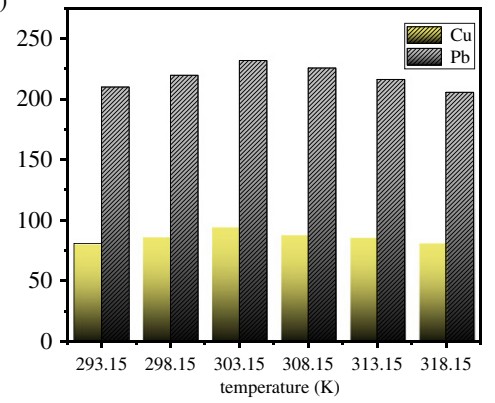

**Figure 9.** Effect of (*a*) pH, (*b*) temperature on the adsorption capacity of Pb(II) and Cu(II).

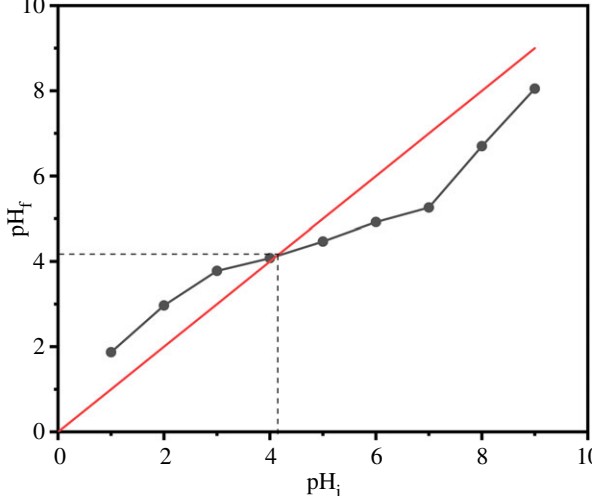

**Figure 10.** Relationship between $pH_{initial}$ and $pH_{final}$ for $pH_{pzc}$ determination.

AO CS-g-PAN. It means that when the pH is 4.13, the total contribution of surface charges is zero. For pH $< pH_{pzc}$, the surface of material had a positive charge. On the other hand, when the initial pH was lower than 4.13, there was a competitive adsorption between heavy ions and $H^+$; it would inhibit the formation of chelating ligands between metal ions and adsorbent. Whereas for pH $> pH_{pzc}$, the surface of material carried a negative charge, the electrostatic repulsion became weakened with the increasing of pH, the adsorption sites became activated and the amounts of adsorbed metal ions increased obviously. It can be determined that the largest adsorption capacities were 231.24 and 94.72 mg g$^{-1}$ at optimum pH 5.0 and 6.0, respectively. However, when the pH exceeds 7, the adsorption capacities declined significantly; it was suspicious to attribute this decrease to the formation of metal hydroxide species [54].

## 3.10. Effect of temperature

The effect of the temperature on the adsorption of Cu(II) and Pb(II) is displayed in figure 9*b*. Initially, the adsorption increased with increasing temperature. When the temperature reached 303.15 K, it performed the maximum adsorption capacities of Cu(II) and Pb(II), but thereafter, the subsequent increase would lead to a decrease in their adsorption capacity. This probably was caused by the thermal energy of metal ion changes with temperature. Initially, thermal energy increased with the rising temperature, giving rise to more probability of binding the metal ions with the vacancies on the adsorbent. But when the thermal vibration of the metal ion was faster than the metal-adsorbent interaction, the metal ion was released back into the solution, resulting in reduced adsorption [55].

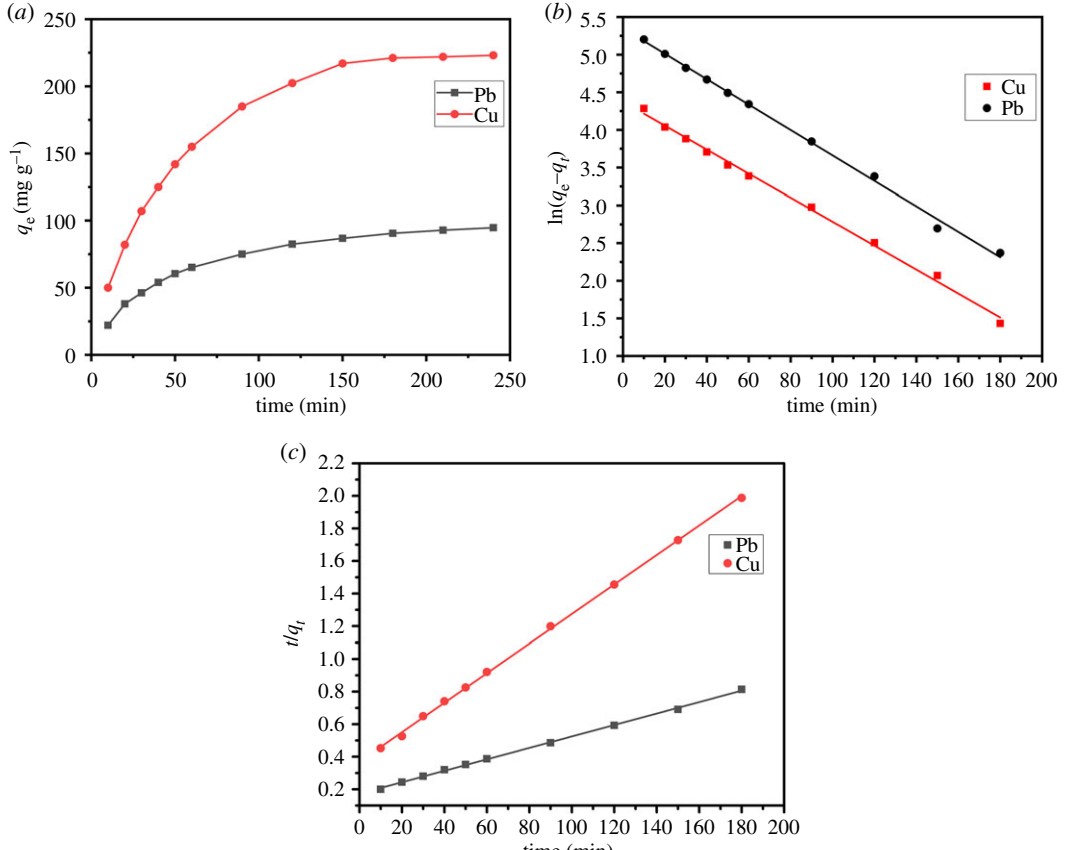

**Figure 11.** (*a*) The adsorption of Pb(II) and Cu(II) onto adsorbent as a function of time, plots of (*b*) pseudo-first order kinetics and (*c*) pseudo-second order kinetics.

## 3.11. Adsorption kinetics

As illustrated in figure 11*a*, the adsorption capacity increased rapidly with the time in the initial period, and then presented a much lower adsorption rate until its attaining the adsorption equilibrium. In order to research the rate-controlling steps and the adsorption procedure of Cu(II) and Pb(II) on the adsorbent, the pseudo-first order model and the pseudo-second order model were used to analyse data of the adsorption kinetics, and the models were defined in the lineal form by using the following equations [56,57]:

$$\ln(q_e - q_t) = \ln q_e - k_1 t \tag{3.1}$$

and

$$\frac{t}{q_t} = \frac{1}{k_2 q_e^2} + \frac{t}{q_e}, \tag{3.2}$$

where $q_e$ is adsorption capacity at the equilibrium, $q_t$ is adsorption capacity at time $t$, $k_1$ and $k_2$ being first order and second order rates, respectively. Relevant parameters calculated according to above kinetic models are summarized in table 3. After comparing the $R^2$ values of the plots obtained, as shown in figure 11*b,c*, and based on the correlation coefficient $R_2^2 = 0.9991$ (Pb) and $R_2^2 = 0.9994$ (Cu), it was found out that the adsorption process conformed better to the pseudo-second order model [58], proving the coexistence of adsorption and chemical reaction during the removal process of Cu(II) and Pb(II) [59].

## 3.12. Adsorption isotherms

In order to illustrate the adsorption relation between the amount of the adsorbed and the solute concentration, the adsorption experimental data were analysed by the Langmuir model (equation

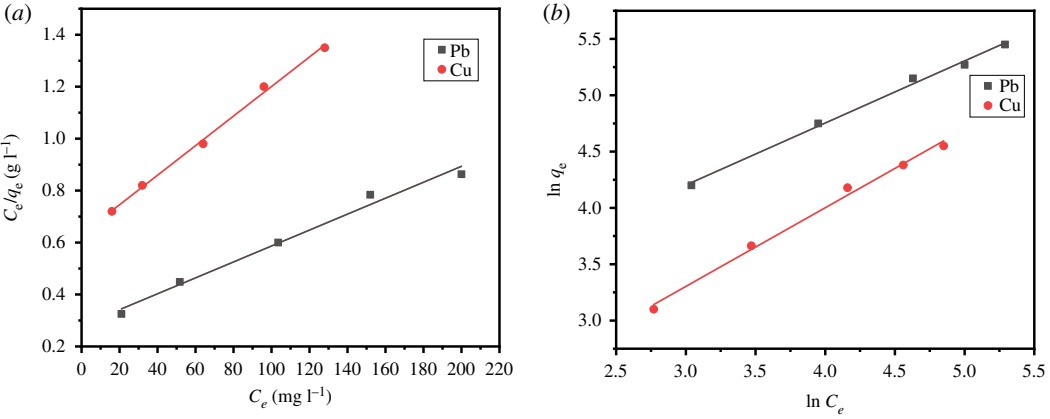

**Figure 12.** Isotherm plots of (*a*) Langmuir and (*b*) Freundlich isotherm.

**Table 3.** Kinetic parameters of pseudo-first order and pseudo-second order models.

| | pseudo-first order | | | pseudo-second order | | |
|---|---|---|---|---|---|---|
| | $k_1$ (mg g$^{-1}$ min$^{-1}$) | $q_m$ (mg g$^{-1}$) | $R_1^2$ | $k_2$ (mg g$^{-1}$ min$^{-1}$) | $q_m$ (mg g$^{-1}$) | $R_2^2$ |
| Pb(II) | 0.0171 | 235.51 | 0.9970 | 0.0005 | 235.85 | 0.9991 |
| Cu(II) | 0.0173 | 96.47 | 0.9963 | 0.00013 | 94.43 | 0.9994 |

**Table 4.** Constants of Langmuir and Freundlich model for Pb(II) and Cu(II) adsorption on AO CS-g-PAN.

| | Langmuir | | | Freundlich | | |
|---|---|---|---|---|---|---|
| | $q_m$ (mg g$^{-1}$) | $b$ (mg l$^{-1}$) | $R^2$ | $k_F$ (mg g$^{-1}$) | $n$ | $R^2$ |
| Pb(II) | 249.19 | 0.0167 | 0.98127 | 10.623 | 1.655 | 0.99328 |
| Cu(II) | 160 | 0.0101 | 0.99567 | 2.382 | 1.241 | 0.99105 |

(3.3)) and the Freundlich model (equation (3.4)), respectively [60].

$$\frac{c_e}{q_e} = \frac{1}{bq_m} + \frac{c_e}{q_m} \tag{3.3}$$

and

$$\ln q_e = \ln k_F + \frac{1}{n}\ln c_e, \tag{3.4}$$

where $q_e$ is the equilibrium adsorption capacity, $q_m$ is the maximum adsorption capacity, $c_e$ is the equilibrium concentration of the metal ions, $b$ is the Langmuir constant related to the binding energy. $k_F$ is the Freundlich constant, and $n$ is the empirical constant which indicates heterogeneity factor. The linear Langmuir isotherm and the Freundlich model are shown in figure 12, and the parameters are given in table 4. Determined with the correlation coefficient $R^2$, for Cu(II), the Freundlich model with $R^2$ value 0.99105 is less suitable for explaining the adsorption process as compared with the Langmuir plot with $R^2$ 0.99567. But for Pb(II), the Freundlich model fitted better as the value of $R^2 = 0.99328$. Both Langmuir and Freundlich isotherms may adequately describe the same set of data for liquid–solid adsorption over a range of concentrations, especially if the solution concentration is small and the adsorption capacity of the solid is large enough to make both isotherm equations approach a linear form [50].

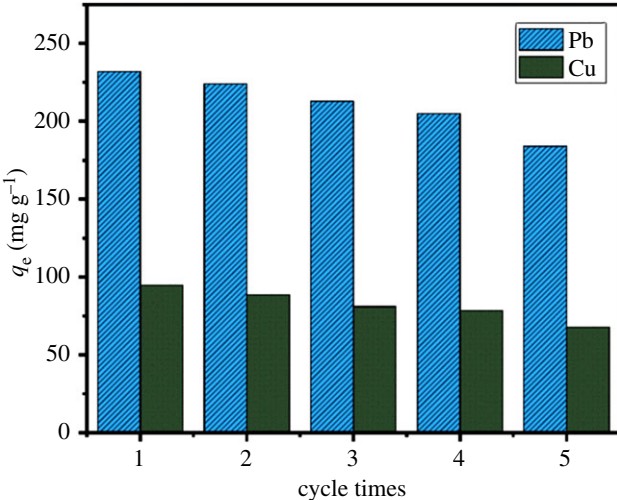

**Figure 13.** Different cycles on the adsorption capacity of Pb(II) and Cu(II) by AO CS-g-PAN.

**Table 5.** Comparison of the adsorption capacities of various adsorbents for adsorption of Pb(II) and Cu(II) ions from wastewater.

| metal ion | adsorbent | $q_m$ (mg g$^{-1}$) | reference |
|---|---|---|---|
| Pb(II) | carboxymethylated cellulose | 24.59 | [63] |
| | chitosan beads | 34.98 | [64] |
| | epichlorohydrin cross-linked chitosan | 34.13 | [65] |
| | chitosan–glutaraldehyde (GLA) beads | 14.24 | [64] |
| | chitosan–GLA–citric acid (C-Gch) flake | 103.6 | [66] |
| Cu(II) | CS-g-PAN | 28.8 | [30] |
| | CS-g-PAM | 19.2 | [30] |

## 3.13. Recycling studies and comparison of adsorption capacity with other literature

The reusability of adsorbents is an important factor for the cost reductions in practical application [22,61]. Several adsorption/desorption cycles were performed to evaluate the stability and reusability of the adsorbent. For regeneration of the adsorbent, the adsorbent loaded with the metal ions was treated with 1 mol l$^{-1}$ HCl solutions and then washed with deionized water, and the process was repeated five times. As shown in figure 13, the adsorbent maintained a satisfying adsorption performance even after five cycles, suggesting that the adsorbent exhibited a predominant reusability. Furthermore, it could be observed that the adsorption capacity was reduced gradually, attributed to the loosing of some of the functional groups of the adsorbent due to the acid cleavage [62]. As shown in table 5, the adsorbent AO CS-g-PAN shows excellent adsorption performance compared with other adsorbents which using natural polymers as raw materials and modified by different methods. Based on the satisfactory adsorption capacity and recyclability, the adsorbent AO CS-g-PAN has great potential to treat aqueous solution pollution effectively.

## 4. Conclusion

In summary, a novel efficient adsorbent AO CS-g-PAN with the coarse surface, loose structure and high adsorption efficiency via SET-LRP and modification process were successfully prepared. FTIR, SEM and XPS confirmed the successful modification of the adsorbent AO CS-g-PAN. The research on the adsorption properties of AO CS-g-PAN has demonstrated that the adsorption was highly pH-dependent. The maximum adsorption was acquired at the amount of adsorbent is 10 mg, when the reaction temperature was set at 30°C. The kinetics studies uncovered that the adsorption process well abided by the pseudo-second order model. The essential adsorption properties were inspected by the

Langmuir model and the Freundlich model, and the results showed that the adsorption of Cu(II) could be explained by the Langmuir isotherm model, while the adsorption of Pb(II) suited the Freundlich isotherm model better. The regeneration experiments suggested the adsorbent was of certain reusability. Based on the above findings, the adsorbent with corn stalk as the raw material can be used extensively as a promising adsorbent in the removal of heavy metal ions from the industrial effluents.

Data accessibility. This article does not contain any additional data.

Authors' contributions. S.L. designed the research, performed the experiments and analysed the data. Y.W. and L.M. analysed and interpreted the data. S.D. and L.L. drafted the manuscript. All authors revised and approved the final form of the manuscript.

Competing interests. We declare we have no competing interests.

Funding. This research was supported by the National Natural Science Foundation of China (grant no. 21376127), the Fundamental Research Funds in Heilongjiang Provincial Universities (Plant food processing technology specialty subject project no. YSTSXK201860), the Fundamental Research Project in Heilongjiang Provincial Education Department (Key projects of science and engineering grant no. 135209102), the Qiqihar City Science and Technology Bureau Project (grant no. GYGY-201601), the Fundamental Research Funds in Heilongjiang Provincial Universities (grant no. 135309110) and the Graduate Innovation Research Project of Qiqihar University (grant no. YJSCX2018-ZD19).

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
