## [Reviewer comments · Royal Society Open Science]

Review History

RSOS-191371.R0 (Original submission)

Review form: Reviewer 1

Is the manuscript scientifically sound in its present form?

No

Are the interpretations and conclusions justified by the results?

No

Is the language acceptable?

Yes

Do you have any ethical concerns with this paper?

No

Have you any concerns about statistical analyses in this paper?

No

Recommendation?

Reject

Comments to the Author(s)

I disagree to call the material used here: as “Green” or “eco-friendly”. During its synthesis, toxic compounds such as DMF and 2-BuBBr are employed since using hazardous and toxic compounds during synthesis is one of the basic concepts of green synthesis. I think authors must state clearly the nature of the material (maybe organic-based) and avoid any green-washing.

Page 5, line 28:

“After graft modification, the content of N element increased from 0% to 18.23%, demonstrating that the acrylonitrile monomer had been grafted onto the surface of the macroinitiator.” Why an increase on the nitrogen content represents indeed that has been grafted? Seems like the authors employed XPS for characterization but they are only reporting weight percentages were is the spectroscopical evidence of grafting?

Surface area and charge distribution are key feature of adsorbent’s characterization but in this study was not determined.

How was the pH monitored and adjusted during adsorption experiments? How the authros decided that equilibrium was effectively reached.

Page 7, line 47: “However, when the pH exceeded to a certain extent, heavy ions and the basic pH adjuster were prone to complexation or precipitation, resulting in a decrease in the adsorption capacity”

This is rather contradictory. If the authors are measuring adsorption capacity from the equilibrium concentration then metal precipitation must indeed decrease the overall concentration in the aqueous phase, giving the wrongful impression that adsorption capacity has been increased. In other words how do the authors know that Q at the highest measured pH indeed is only adsorption?

Review form: Reviewer 2 (Ismat Ali)**Is the manuscript scientifically sound in its present form?**

Yes

Are the interpretations and conclusions justified by the results?

No

Is the language acceptable?

Yes

Do you have any ethical concerns with this paper?

No

Have you any concerns about statistical analyses in this paper?

No

Recommendation?

Major revision is needed (please make suggestions in comments)

Comments to the Author(s)

Summary section should not be numbered (see Appendix A).

Decision letter (RSOS-191371.R0)

02-Sep-2019

Dear Mrs Wang:

Manuscript ID: RSOS-191371

Title: "Corn stalk as starting material to prepare a green adsorbent via SET-LRP and its adsorption performance for Pb(II) and Cu(II)"

Thank you for submitting the above manuscript to Royal Society Open Science. Your paper was sent to reviewers and their comments are included at the bottom of this letter.

In view of the concerns raised by the reviewers, the manuscript has been rejected in its current form. However, a new manuscript may be submitted which takes into consideration these comments.

Please note that resubmitting your manuscript does not guarantee eventual acceptance, and that your resubmission will be subject to peer review before a decision is made.

Your resubmitted manuscript should be submitted by 01-Mar-2020. If you are unable to submit by this date please contact the Editorial Office.

On behalf of the Subject Editor Professor Anthony Stace and the Associate Editor Dr Ya-Wen Wang

REVIEWER(S) REPORTS:
Associate Editor Comments to Author ():
RSC Associate Editor:
Comments to the Author:
(There are no comments.)

RSC Subject Editor:

Comments to the Author:
(There are no comments.)

Reviewers' Comments to Author:
Reviewer: 1

Comments to the Author(s)

I disagree to call the material used here: as "Green" or "eco-friendly". During its synthesis, toxic compounds such as DMF and 2-BuBBr are employed since using hazardous and toxic compounds during synthesis is one of the basic concepts of green synthesis. I think authors must state clearly the nature of the material (maybe organic-based) and avoid any green-washing.

Page 5, line 28:

"After graft modification, the content of N element increased from 0% to 18.23%, demonstrating that the acrylonitrile monomer had been grafted onto the surface of the macroinitiator." Why an increase on the nitrogen content represents indeed that has been grafted? Seems like the authors employed XPS for characterization but they are only reporting weight percentages were is the spectroscopical evidence of grafting?

Surface area and charge distribution are key feature of adsorbent's characterization but in this study was not determined.

How was the pH monitored and adjusted during adsorption experiments? How the authros decided that equilibrium was effectively reached.

Page 7, line 47: "However, when the pH exceeded to a certain extent, heavy ions and the basic pH adjuster were prone to complexation or precipitation, resulting in a decrease in the adsorption capacity"

This is rather contradictory. If the authors are measuring adsorption capacity from the equilibrium concentration then metal precipitation must indeed decrease the overall concentration in the aqueous phase, giving the wrongful impression that adsorption capacity has been increased. In other words how do the authors know that Q at the highest measured pH indeed is only adsorption?

Reviewer: 2

Comments to the Author(s)
Summary section should not be numbered.

Author's Response to Decision Letter for (RSOS-191371.R0)

See Appendix B.

RSOS-191811.R0

Review form: Reviewer 1

Is the manuscript scientifically sound in its present form?

No

Are the interpretations and conclusions justified by the results?

No

Is the language acceptable?

Yes

Do you have any ethical concerns with this paper?

No

Have you any concerns about statistical analyses in this paper?

No

Recommendation?

Reject

Comments to the Author(s)

After reviewing carefully the manuscript I think is not suitable for publication, few reasons follows:

"ecofriendly" still appears through the text.

It is irreal to speculate about porosity in such low porous materials, by the way significant figures in surface area measurements make not sense.

It is imposible to draw conclusion about adsorption mechanisms based merely in the trends followed by the data of the adsorption isotherm, particularly when both have a high R2 value.

There are not single evidence of such adsorption mechanisms.

Page 6, line 59: "good adsorptivity" how come? I think the authors should put attention in the difference of adsorption vs absorption.

I do not follow the word "sabotage" in line 60 of page 6.

Review form: Reviewer 2 (Ismat Ali)

Is the manuscript scientifically sound in its present form?

Yes

Are the interpretations and conclusions justified by the results?

Yes

Is the language acceptable?

Yes

Do you have any ethical concerns with this paper?

No

Have you any concerns about statistical analyses in this paper?

No

Recommendation?

Accept with minor revision (please list in comments)

Comments to the Author(s)

The authors made most of the necessary arrangements considering most of the warnings but still some points should be clarified.

- 1) The circumstances of adsorption experiments should be mentioned e.g. did authors use thermostated water bath shaker or magnetic stirrer? At what rpm? At what temperature?
- 2) The removal efficiency % for both metals should be calculated and added to the abstract and results.

- 3) Batch Adsorption Studies: How much metal is lost by adsorption to the walls of the glass flask or the magnetic stirrer? A control without adsorbent should be used.
- 4) What are the initial concentrations of both metals used in this study?
- 5) There is no true control used in this work. It would be useful to see the data from the untreated Corn stalk as a comparison.
- 6) A very common error in sorption studies is the absence of replicates. How many times were the experiments performed?
- 7) It would be more useful to present a table that demonstrates some key parameters (e.g. maximum adsorption capacity) from the previous literature to help make a case that your work is necessary.
- 8) Although the authors studied the temperature effect but the thermodynamic parameters were not given. All thermodynamic parameters should be calculated, tabulated and discussed in details.
- 9) The arrangement of section 3 (results and discussion) is not suitable. All characterization measurements should be reported first and then followed by adsorption results i.e. Surface area and pore size distribution, XPS spectrum analysis and EDS spectrum analysis should be reported directly after Elemental analysis.

Review form: Reviewer 3

Is the manuscript scientifically sound in its present form?

Yes

Are the interpretations and conclusions justified by the results?

No

Is the language acceptable?

No

Do you have any ethical concerns with this paper?

Yes

Have you any concerns about statistical analyses in this paper?

No

Recommendation?

Major revision is needed (please make suggestions in comments)

Comments to the Author(s)

- (1) I do not agree the description "green adsorbent" or "eco-friendly", because almost all the raw materials used are chemicals and even petroleum-based monomers, except CS.
- (2) English needs to be improved. too many errors can be found, such as "pH drift method"; is "pore structures" "pore structure parameters"? how do you determine "pore structure"?
- (3) The introduction on Corn stalk is not enough. More information about it should be provided, such as the main chemical composition, ingredients, and so on.
- (4) DTG and DSC curves are required to obtain T_{max} and reveal the maximum thermal decomposition position.
- (5) The effect of graft ration on adsorption capacity needs to be studied.
- (6) A comparison of the adsorbent with other reported adsorbents needs to be made.

Review form: Reviewer 4

Is the manuscript scientifically sound in its present form?

Yes

Are the interpretations and conclusions justified by the results?

Yes

Is the language acceptable?

Yes

Do you have any ethical concerns with this paper?

No

Have you any concerns about statistical analyses in this paper?

No

Recommendation?

Major revision is needed (please make suggestions in comments)

Comments to the Author(s)

This study provides an alternative way for adsorbing Pb(II) and Cu(II) by Corn stalk as starting material. The adsorbent exhibited a predominant adsorption performance on Pb(II) and Cu(II). However, the following concerns should be further considered and addressed.

1. The material used here: as “eco-friendly” or “Green” , for instance, line 1 and line 4 in summary. Please check all over the manuscript. If you insist your own opinion, you should compare your material with others to prove your material is “eco-friendly” or “Green”.
2. The manuscript didn't show CS, CS-Br and CS-g-PAN on the adsorption capacity of Pb(II) and Cu(II) to compare their capacity with AO CS-g-PAN.
3. The introduction about adsorption mechanism is not enough, and the authors are suggested to cite this papers:

□ Junqin Liu, Pingxiao Wu, ShanShan Yang, Saeed Rehman, ZubairAhmed, Nengwu Zhu, Zhi Dang, Zehua Liu. A photo-switch for peroxydisulfatenon-radical/radical activation over layered CuFe oxide: Rational degradation pathway choice for pollutants. Applied Catalysis B: Environmental, 2020, 261: 118232.

□ Junqin Liu, Pingxiao Wu, Shuaishuai Li, Meiqing Chen, Wentin Cai, Dinghui Zou, Nengwu Zhu, Zhi Dang. Synergistic deep removal of As(III) and Cd(II) by a calcined multifunctional MgZnFe-CO₃ layered double hydroxide: Photooxidation, precipitation and adsorption. Chemosphere, 2019, 225: 115-125.

□ Shanshan Yang, Zhiyan Huang, Pingxiao Wu, Yihao Li, Xiongbo Dong, Chunquan Li, Ningyuan Zhu, Xiaodi Duan, Dionysios D. Dionysiou. Rapid removal of tetrabromobisphenol A by α -Fe₂O₃-x@Graphene@Montmorillonite catalyst with oxygen vacancies through peroxymonosulfate activation: Role of halogen and α -hydroxyalkyl radicals. Applied Catalysis B: Environmental, 2020, 260: 118129.

□ Meiqing Chen, Pingxiao Wu, Langfeng Yu, Shuai Liu, Bo Ruan, Haihui Hu, Nengwu Zhu, Zhang Lin. FeOOH-loaded MnO₂ nano-composite: An efficient emergency material for thallium pollution incident. Journal of Environmental Management, 2017, 192: 31-38.

□ Shanshan Yang, Pingxiao Wu, Junqin Liu, Meiqing Chen, Zubair Ahmed, Nengwu Zhu. Efficient removal of bisphenol A by superoxide radical and singlet oxygen generated from peroxymonosulfate activated with Fe⁰-montmorillonite. Chemical Engineering Journal, 2018, 350: 484-495.

□ Liya Chen, Pingxiao Wu, Meiqing Chen, Xiaolin Lai, Zubair Ahmed, Nengwu Zhu, Zhi Dang, Yingzhi Bi, Tongyun Liu. Preparation and characterization of the eco-friendly chitosan/vermiculite biocomposite with excellent removal capacity for cadmium and lead. Applied Clay Science, 2018,159: 74-82.

- Shuai Liu, Pingxiao Wu, Meiqing Chen, Langfeng Yu, Chunxi Kang, Nengwu Zhu, Zhi Dang. Amphoteric modified vermiculites as adsorbents for enhancing removal of organic pollutants: Bisphenol A and Tetrabromobisphenol A. *Environmental Pollution*, 2017, 228: 277-286.
- Chongmin Liu, Pingxiao Wu, Yajie Zhu, Lytuong Tran. Simultaneous adsorption of Cd²⁺ and BPA on amphoteric surfactant activated montmorillonite. *Chemosphere*, 2016, 144: 1026-1032.

Review form: Reviewer 5

Is the manuscript scientifically sound in its present form?

No

Are the interpretations and conclusions justified by the results?

Yes

Is the language acceptable?

No

Do you have any ethical concerns with this paper?

No

Have you any concerns about statistical analyses in this paper?

No

Recommendation?

Major revision is needed (please make suggestions in comments)

Comments to the Author(s)

1. I agree with Reviewer 1 that the materials prepared in this study are not "green" or "eco-friendly". The authors made correction but there are still some left in the manuscript: "eco-friendly" in Summary and Line 59, P9 and "green adsorbent" in Summary.

2. There are some grammar errors and typos in the manuscript. Please find a native speaker to check and rephrase if possible.

For an example, "including the" repeated twice in Line 7, P 11.

And what is "SET-LRP"? The full name of this abbreviation never appeared.

3. Materials and Methods

XPS is used to detect the surface composition. For a newly synthesized organic/polymeric material, elemental analysis is usually necessary.

What are the initial concentrations of the adsorbates in batch adsorption tests?

4. Results and Discussion

The BET surface area of the obtained material was < 1 m²/g and the total pore volume around 0.1 cm³/g, which implied the small surface and almost nonporous nature of the material. In this case it is questionable whether this material is an appropriate candidate for adsorbent.

“Thermal degradation behaviors analysis” is not supported by relevant literature. Give the references please.

The adsorption isotherm (q_e vs C_e) in Figure 11 looked more like a linear partition of solute on the material rather than adsorption. And the units are absent in Figure 11(a). How did you determine the number of significant figure in Table 3 and 4? Is 5 correct and reasonable?

What are the equilibrium concentrations of the adsorbate corresponding to the equilibrium (maximum) adsorption capacity 231.84 mg/g and 94.72 mg/g respectively? It won't make sense to compare the adsorption capacity without mentioning the equilibrium concentration.

The "Effect of the dosage of adsorbent" part is unnecessary and superfluous now that the adsorption isotherms are obtained.

Decision letter (RSOS-191811.R0)

06-Jan-2020

Dear Mrs Wang:

Title: Corn stalk as starting material to prepare a novel adsorbent via SET-LRP and its adsorption performance for Pb(II) and Cu(II)
Manuscript ID: RSOS-191811

The editor assigned to your paper has now received comments from reviewers. We would like you to revise your paper in accordance with the referee and Subject Editor suggestions which can be found below (not including confidential reports to the Editor). Please note this decision does not guarantee eventual acceptance.

Please submit a copy of your revised paper before 29-Jan-2020. Please note that the revision deadline will expire at 00.00am on this date. If we do not hear from you within this time then it will be assumed that the paper has been withdrawn. In exceptional circumstances, extensions may be possible if agreed with the Editorial Office in advance. We do not allow multiple rounds of revision so we urge you to make every effort to fully address all of the comments at this stage. If deemed necessary by the Editors, your manuscript will be sent back to one or more of the original reviewers for assessment. If the original reviewers are not available we may invite new reviewers.

Once again, thank you for submitting your manuscript to Royal Society Open Science and I look

forward to receiving your revision. If you have any questions at all, please do not hesitate to get in touch.

RSC Associate Editor
Comments to the Author:
According to the comments of three adjudicators, the decision was made.

Reviewers' Comments to Author:
Reviewer: 1

Comments to the Author(s)
After reviewing carefully the manuscript I think is not suitable for publication, few reasons follows:
"ecofriendly" still appears through the text.
It is irreal to speculate about porosity in such low porous materials, by the way significant figures in surface area measurements make not sense.
It is imposible to draw conclusion about adsorption mechanisms based merely in the trends followed by the data of the adsorpton isotherm, particularly when both have a high R2 value. There are not single evidence of such adsorption mechanisms.
Page 6, line 59: "good absopitivity" how come? I think the authors should put attention in the difference of adsorption vs absorption.
I do not follow the word "sabotage" in line 60 of page 6.

Reviewer: 2

Comments to the Author(s)
The authors made most of the necessary arrangements considering most of the warnings but still some points should be clarified.
1) The circumstances of adsorption experiments should be mentioned e.g. did authors use thermostated water bath shaker or magnetic stirrer? At what rpm? At what temperature?
2) The removal efficiency % for both metals should be calculated and added to the abstract and results.
3) Batch Adsorption Studies: How much metal is lost by adsorption to the walls of the glass flask or the magnetic stirrer? A control without adsorbent should be used.
4) What are the initial concentrations of both metals used in this study?
5) There is no true control used in this work. It would be useful to see the data from the untreated Corn stalk as a comparison.
6) A very common error in sorption studies is the absence of replicates. How many times were the experiments performed?

- 7) It would be more useful to present a table that demonstrates some key parameters (e.g. maximum adsorption capacity) from the previous literature to help make a case that your work is necessary.
- 8) Although the authors studied the temperature effect but the thermodynamic parameters were not given. All thermodynamic parameters should be calculated, tabulated and discussed in details.
- 9) The arrangement of section 3 (results and discussion) is not suitable. All characterization measurements should be reported first and then followed by adsorption results i.e. Surface area and pore size distribution, XPS spectrum analysis and EDS spectrum analysis should be reported directly after Elemental analysis.

Reviewer: 3

Comments to the Author(s)

- (1) I do not agree the description "green adsorbent" or "eco-friendly", because almost all the raw materials used are chemicals and even petroleum-based monomers, except CS.
- (2) English needs to be improved. too many errors can be found, such as "pH drift method"; is "pore structures" "pore structure parameters"? how do you determine "pore structure"?
- (3) The introduction on Corn stalk is not enough. More information about it should be provided, such as the main chemical composition, ingredients, and so on.
- (4) DTG and DSC curves are required to obtain T_{max} and reveal the maximum thermal decomposition position.
- (5) The effect of graft ration on adsorption capacity needs to be studied.
- (6) A comparison of the adsorbent with other reported adsorbents needs to be made.

Reviewer: 4

Comments to the Author(s)

This study provides an alternative way for adsorbing Pb(II) and Cu(II) by Corn stalk as starting material. The adsorbent exhibited a predominant adsorption performance on Pb(II) and Cu(II). However, the following concerns should be further considered and addressed.

1. The material used here: as "eco-friendly" or "Green", for instance, line 1 and line 4 in summary. Please check all over the manuscript. If you insist your own opinion, you should compare your material with others to prove your material is "eco-friendly" or "Green".
2. The manuscript didn't show CS, CS-Br and CS-g-PAN on the adsorption capacity of Pb(II) and Cu(II) to compare their capacity with AO CS-g-PAN.
3. The introduction about adsorption mechanism is not enough, and the authors are suggested to cite this papers:
 - Junqin Liu, Pingxiao Wu, ShanShan Yang, Saeed Rehman, ZubairAhmed, Nengwu Zhu, Zhi Dang, Zehua Liu. A photo-switch for peroxydisulfate non-radical/radical activation over layered CuFe oxide: Rational degradation pathway choice for pollutants. Applied Catalysis B: Environmental, 2020, 261: 118232.
 - Junqin Liu, Pingxiao Wu, Shuaishuai Li, Meiqing Chen, Wentin Cai, Dinghui Zou, Nengwu Zhu, Zhi Dang. Synergistic deep removal of As(III) and Cd(II) by a calcined multifunctional MgZnFe-CO₃ layered double hydroxide: Photooxidation, precipitation and adsorption. Chemosphere, 2019, 225: 115-125.
 - Shanshan Yang, Zhiyan Huang, Pingxiao Wu, Yihao Li, Xiongbo Dong, Chunquan Li, Ningyuan Zhu, Xiaodi Duan, Dionysios D. Dionysiou. Rapid removal of tetrabromobisphenol A by α -Fe₂O₃-x@Graphene@Montmorillonite catalyst with oxygen vacancies through peroxymonosulfate activation: Role of halogen and α -hydroxyalkyl radicals. Applied Catalysis B: Environmental, 2020, 260: 118129.
 - Meiqing Chen, Pingxiao Wu, Langfeng Yu, Shuai Liu, Bo Ruan, Haihui Hu, Nengwu Zhu, Zhang Lin. FeOOH-loaded MnO₂ nano-composite: An efficient emergency material for thallium pollution incident. Journal of Environmental Management, 2017, 192: 31-38.

- Shanshan Yang, Pingxiao Wu, Junqin Liu, Meiqing Chen, Zubair Ahmed, Nengwu Zhu. Efficient removal of bisphenol A by superoxide radical and singlet oxygen generated from peroxymonosulfate activated with Fe⁰-montmorillonite. *Chemical Engineering Journal*, 2018, 350: 484-495.
- Liya Chen, Pingxiao Wu, Meiqing Chen, Xiaolin Lai, Zubair Ahmed, Nengwu Zhu, Zhi Dang, Yingzhi Bi, Tongyun Liu. Preparation and characterization of the eco-friendly chitosan/vermiculite biocomposite with excellent removal capacity for cadmium and lead. *Applied Clay Science*, 2018, 159: 74-82.
- Shuai Liu, Pingxiao Wu, Meiqing Chen, Langfeng Yu, Chunxi Kang, Nengwu Zhu, Zhi Dang. Amphoteric modified vermiculites as adsorbents for enhancing removal of organic pollutants: Bisphenol A and Tetrabromobisphenol A. *Environmental Pollution*, 2017, 228: 277-286.
- Chongmin Liu, Pingxiao Wu, Yajie Zhu, Lytuong Tran. Simultaneous adsorption of Cd²⁺ and BPA on amphoteric surfactant activated montmorillonite. *Chemosphere*, 2016, 144: 1026-1032.

Reviewer: 5

Comments to the Author(s)

1. I agree with Reviewer 1 that the materials prepared in this study are not "green" or "eco-friendly". The authors made correction but there are still some left in the manuscript: "eco-friendly" in Summary and Line 59, P9 and "green adsorbent" in Summary.

2. There are some grammar errors and typos in the manuscript. Please find a native speaker to check and rephrase if possible.

For an example, "including the" repeated twice in Line 7, P 11.

And what is "SET-LRP"? The full name of this abbreviation never appeared.

3. Materials and Methods

XPS is used to detect the surface composition. For a newly synthesized organic/polymeric material, elemental analysis is usually necessary.

What are the initial concentrations of the adsorbates in batch adsorption tests?

4. Results and Discussion

The BET surface area of the obtained material was < 1 m²/g and the total pore volume around 0.1 cm³/g, which implied the small surface and almost nonporous nature of the material. In this case it is questionable whether this material is an appropriate candidate for adsorbent.

"Thermal degradation behaviors analysis" is not supported by relevant literature. Give the references please.

The adsorption isotherm (q_e vs C_e) in Figure 11 looked more like a linear partition of solute on the material rather than adsorption. And the units are absent in Figure 11(a). How did you determine the number of significant figure in Table 3 and 4? Is 5 correct and reasonable?

What are the equilibrium concentrations of the adsorbate corresponding to the equilibrium (maximum) adsorption capacity 231.84 mg/g and 94.72 mg/g respectively? It won't make sense to compare the adsorption capacity without mentioning the equilibrium concentration.

The "Effect of the dosage of adsorbent" part is unnecessary and superfluous now that the adsorption isotherms are obtained.

Author's Response to Decision Letter for (RSOS-191811.R0)

See Appendix C.

Decision letter (RSOS-191811.R1)

04-Feb-2020

Dear Mrs Wang:

Title: Corn stalk as starting material to prepare a novel adsorbent via SET-LRP and its adsorption performance for Pb(II) and Cu(II)
Manuscript ID: RSOS-191811.R1

It is a pleasure to accept your manuscript in its current form for publication in Royal Society Open Science. The chemistry content of Royal Society Open Science is published in collaboration with the Royal Society of Chemistry.

RSC Associate Editor
Comments to the Author:
(There are no comments.)

Reviewer(s)' Comments to Author:

Appendix A

This paper presented an adsorbent prepared from corn stalk for heavy metal adsorption from wastewater. Please consider the following:

The manuscript can be accepted for publication if the authors perform the following major revisions.

- 1) The percentage of removal efficiency for both metals should be calculated and added to the abstract and results as well.
- 2) The background (Introduction) needs polishing. What specific wastewaters have lead and copper as primary constituents of concern?
- 3) Authors describe their adsorbent as (green adsorbent), I suggest to delete (green adsorbent) from the title of the manuscript because many chemicals were used to synthesize the adsorbent, so it cannot be described as green adsorbent.
- 4) Your paragraph starting on page 3 line 1 needs to be reconsidered. Some of the biomasses listed were converted to activated carbon and then applied to adsorption of heavy metals. It would be more useful to present a table that demonstrates some key parameters (e.g. maximum adsorption capacity) from the previous literature to help make a case that your work is necessary.
- 5) Your experimental approach appears sound, but there are a few pieces of information that would be useful in making your case. First, how does ionic strength of the liquid influence the adsorption? Wastewaters do not contain only the heavy metals. Second, how interfering heavy metals influence on the removal efficiency?
- 6) While you have identified a viable adsorbent, you have not made an effective argument that your adsorbent was needed or that it is superior to existing adsorbents discussed in literature. The first line of your conclusion could be strengthened if you compare your results to those previously reported in the literature (see comment 4).

- 7) Authors stated that they studied the effect of pH on the adsorption efficiency over the range 2-7. The effect of pH should be studied over the basic pH range as well.
- 8) Some parameters obtained from Langmuir and Freundlich models are presented in Table 3. However, more discussion is needed. What do b , n and K_f values mean?
- 9) Although the authors studied the temperature effect but the thermodynamic parameters were not given. All thermodynamic parameters should be calculated, tabulated and discussed in details.
- 10) What is initial concentration used in this study?
- 11) Page 4 line 39 the word(final) should be changed to equilibrium or remaining.
- 12) The adsorption nature should be determined (physical or chemical)
- 13) More details on the adsorption experiments are needed such as how experiments were performed (by using thermostated shaker or magnetic stirrer. What is the rpm?)
- 14) Blank experiments (without the adsorbent) should be performed to check is there any amount of heavy metals can be removed by glass walls of the container or by magnetic bar (if used).

Appendix B

Dear Editors and Reviewers,

Thank you for your letter and constructive comments which very helpful for improving our paper. We have studied the comments carefully and have made some corrections in our article. The corrections and the response to reviewer's comments are as following:

Responds to the reviewer's comments:

Reviewer: 1

Comment 1. I disagree to call the material used here: as "Green" or "eco-friendly". During its synthesis, toxic compounds such as DMF and 2-BiBBr are employed since using hazardous and toxic compounds during synthesis is one of the basic concepts of green synthesis. I think authors must state clearly the nature of the material (maybe organic-based) and avoid any green-washing.

Answer: Thank you for your constructive advice. We show our most sincere apology for the inaccurate expressions such as "green" or "eco-friendly", we have made some corrections in the article, and marked the changes in red. It is really true as you suggested, according to the concept of green synthesis, that it is necessary to avoid using toxic and hazardous compounds during synthesis. However, DMF and 2-BiBBr are quite significant in our study due to their vital roles in the preparation of macroinitiator and single electron transfer living radical polymerization (SET-LRP). 2-BiBBr is commonly regarded as an intermediate in organic synthesis. In our synthesis, the 2-BiBBr was employed to prepare a macroinitiator. For the two bromine atoms of BIBB, one bromine atom reacts with the hydroxyl groups on the cellulose to immobilize 2-BiBBr onto the cellulose, and the other bromine atom acted as an active center to initiate the subsequent SET-LRP. Based on the above, BIBB is necessary in our synthesis process because it provided the initiation site for SET-LRP. DMF was applied as the solvent for polymerization system because that the acrylonitrile could dissolve in DMF while the cellulose-g-polyacrylonitrile is insoluble in DMF. Considering that the difference in solubility is beneficial to remove the unreacted acrylonitrile monomer after polymerization, the use of DMF is considered reasonable.

Comment 2. Page 5, line 28:

"After graft modification, the content of N element increased from 0% to 18.23%, demonstrating that the acrylonitrile monomer had been grafted onto the surface of the macroinitiator." Why an increase on the nitrogen content represents indeed that has been grafted? Seems like the authors employed XPS for characterization but they are only reporting weight percentages were is the spectroscopical evidence of grafting?

Answer: Thank you for your valuable comment. In our study, the success of graft modification was jointly demonstrated by FTIR (Figure X) and XPS. We determined that the increase in content of N element measured by XPS and the characteristic peak of cyano at 2242 cm^{-1} measured by FTIR were all attributed to polyacrylonitrile grafted onto the macroinitiator rather than unreacted acrylonitrile monomer or the homopolymer of acrylonitrile. The reasons are as follows. The solvent for the grafting reaction is DMF, the AN monomer was dissolved in DMF while the CS-g-PAN is insoluble. After the grafting, the unreacted AN monomer could be completely removed with the solvent. Moreover, only the macroinitiator is used to initiate the polymerization in the grafting process without a free initiator, so that no homopolymer of acrylonitrile is produced in the system. Based on the above, we can determine that the increase in the N element content and the peak at

2242 cm⁻¹ in the infrared spectrum are indeed attributed to the CS-g-PAN rather than the AN monomer or pure PAN. In addition, the changes of the morphology of the samples before and after grafting shown by SEM can also assist in proving the success of graft modification.

Comment 3. Surface area and charge distribution are key feature of adsorbent's characterization but in this study was not determined.

Answer: Considering the Reviewer's suggestion, we have measured the charge distribution and the surface area of the adsorbent. To determine the charge distribution of the adsorbent, the pH drift method was used to investigate the point of zero surface charge (pH_{pzc}). Surface area and porosity of CS and AO CS-g-PAN were determined by nitrogen adsorption/desorption method. The details and results were description as following, and these paragraphs and Table 2 were added to the revised manuscript and marked in red.

The paragraphs of

“The point of the zero charge (pH_{pzc}) is a very significant indicator which plays a vital role in adsorption phenomena [42]. In order to determine the pH_{pzc}, the pH value in which the electrical charge density of the adsorbent surface is zero, 20 mg of adsorbent were stirred with 20 mL of 0.1 mol L⁻¹ NaNO₃ solution, HNO₃ or NaOH were added to the mixture to adjust the pH (pH₀) values ranging from 1 up to 9, the suspensions were stirred for 48 h. The final pH (pH_f) of suspensions was measured and the plot pH_{final} and pH_{initial} was obtained.”

“Effect of initial pH on adsorption of Cu(II) and Pb(II) was explored in the pH range from 2 to 7, and the variation tendencies of the adsorption capacity of the AO CS-g-PAN at different pH were exhibited in Figure 4a. It illustrated that the adsorption process was pH dependent. The pH_{pzc} was defined as the pH value in which the final pH value equals the initial pH value. As shown in Fig 5, the pH_{pzc} was obtained at the intersection of fitting line and straight line. Therefore, the pH_{pzc} is 4.13 for adsorbent AO CS-g-PAN. It means that when the pH is 4.13, the total contribution of surface charges is zero. For pH < pH_{pzc}, the surface of material had positive charge. On the other word, when the initial pH was lower than 4.13, there was a competitive adsorption between heavy ions and heavy ions and H⁺, it would inhibit the formation of chelating ligands between metal ions and adsorbent. Whereas for pH > pH_{pzc}, the surface of material carried negative charge, the electrostatic repulsion became weaken with the increasing of pH, the adsorption sites became activated and the amounts of adsorbed metal ions increased obviously. It can be determined that the largest adsorption capacities were 231.24 mg/g and 94.72 mg/g at optimum pH 5.0 and 6.0 respectively. However, when the pH exceeds 7, the adsorption capacities declined significantly, it was suspicious to attribute this decrease to the formation of metal hydroxide species. [43]”

“Surface area and porosity of CS and AO CS-g-PAN were determined from the nitrogen adsorption/desorption measurement at 77K with an Autosorb-iQ surface analyzer (Quantachrome Instruments, USA). Samples was degassed at 120 °C to remove the guests. The related important parameters were summary in Table 1. It could be found that the graft modification actually decreased the specific surface area and the pore volume due to the blockage of internal porosity by grafted chain of PAN. It was also observed that graft modification resulted in an increase in average pore diameter, indicating that the blocked pores would be micropores [45]. For the pure physical

adsorption process, the adsorption capacity increases with increasing surface area of the adsorbent [46]. Note that, the untreated corn stalk was also investigated as adsorbent for comparison in our initial adsorption performance test. Results suggest that the maximum adsorption capacity of AO CS-g-PAN on Pb(II) and Cu(II) were 231.84 mg/g and 94.72 mg/g respectively, which is much higher than CS with the maximum adsorption capacity 47.63 mg/g and 19.65 mg/g. Therefore it is speculated that the physisorption of AO CS-g-PAN for Pb(II) and Cu(II) is restricted and the chemisorption is the predominant adsorption mechanism due to the existence of adsorption sites amidoxime groups which effectively bind heavy metal ions.” were added to revised manuscript.

Table 2 Pore structure parameters of CS and AO CS-g-PAN

Sample	S_{BET} (m ² /g)	V_{tot} (cm ³ /g)	Pore width (nm)
CS	2.813	0.084	15.044
AO CS-g-PAN	0.72	0.013	30.767

Comment 4. How was the pH monitored and adjusted during adsorption experiments? How the authors decided that equilibrium was effectively reached.

Answer: We are sorry for the negligence of the lack of description of pH measurement and adjustment. In fact, the pH was determined by portable precise acidimeter pH-100 meter. And the pH was adjusted with 0.1-1mol/L HNO₃ and NaOH solution. In the study of adsorption kinetics, we found that at the initial adsorption, the adsorption capacity increased with increasing time, but after 4 hours, the adsorption capacity hardly increased. It means that the adsorption is almost saturated. For chemisorption, equilibrium adsorption can be considered as saturated adsorption. In our adsorption experiment, the contact time was 8 h when studying the influence of pH, the concentration of metal ions and the dosage of adsorbent on adsorption, except that adsorption kinetics required different adsorption time. This is a sufficient time to achieve saturated adsorption, so it is reasonable to believe that an effective adsorption equilibrium has been reached.

Comment 5. Page 7, line 47: “However, when the pH exceeded to a certain extent, heavy ions and the basic pH adjuster were prone to complexation or precipitation, resulting in a decrease in the adsorption capacity”

This is rather contradictory. If the authors are measuring adsorption capacity from the equilibrium concentration then metal precipitation must indeed decrease the overall concentration in the aqueous phase, giving the wrongful impression that adsorption capacity has been increased. In other words how do the authors know that Q at the highest measured pH indeed is only adsorption?

Answer: Thank you for your helpful comment. We reconsidered this question seriously and found that the complexation or precipitation of heavy ions and the basic pH adjuster does cause a false impression of increased adsorption capacity. The statement of “However, when the pH exceeded to a certain extent, heavy ions and the basic pH adjuster were prone to complexation or precipitation, resulting in a decrease in the adsorption capacity” were corrected as “However, when the pH exceeds 7, the adsorption capacities declined significantly, it was suspicious to attribute this decrease to the formation of metal hydroxide species”. And we will try to find out the reason in the future research.

Reviewer: 2

Comment 1. Summary section should not be numbered.

Answer: We are very sorry for our incorrect numbering, we have corrected the mistake and we will pay more attention to such details in future work.

We have tried our best to improve the paper, and the changes and added content were marked in red in revised manuscript. We appreciate for reviewers' warm work earnestly, and hope that these corrections will meet with approval.

Once again, we appreciate for your valuable comments and suggestions.

Appendix C

Dear editor and reviewers,

Thank you for handling our manuscript (Manuscript ID RSOS-191811). We feel that the editor's and reviewers' comments and suggestions on our paper were very helpful for improving our paper.

We have revised our manuscript carefully according to those comments and suggestions, and a response letter is attached to address all the comments from the referees and the editors. The detailed changes in the revised manuscript were marked in red. Please check the uploaded files of the revised manuscript.

We do appreciate your great efforts on the publication of our work in your respected journal. If you need any additional information, please contact me at any moment.

Thank you very much.

Sincerely yours,

Ya-zhen Wang

Qiqihar University, Qiqihar, P.R.China.

Responds to the reviewer's comments:

Reviewer: 1

Comments to the Author(s)

After reviewing carefully the manuscript I think is not suitable for publication, few reasons follows:

"eco-friendly" still appears through the text.

Answer: Thank you for your valuable comment. We sincerely apologize for the incorrect description such as "green" or "eco-friendly", we have corrected our manuscript carefully, and the changes were marked in red. We have to extend sincere apologies again. We will take every detail seriously and rigorously in our work in the future.

It is unreal to speculate about porosity in such low porous materials, by the way significant figures in surface area measurements make no sense.

Answer: Thank you very much for pointing out the valuable comment. Based on the pore structure parameters were determined from the nitrogen adsorption/desorption measurement, the material does have a small surface and almost nonporous nature. But we want to make it clear that the adsorbent we prepared mainly chemical adsorption removal of metal ions, rather than physical adsorption, it can be proved by the adsorption mechanism, namely the formation of new chemical bonds between metal ions and adsorbents. Furthermore, the result was consistent with previous research reported by Gedam [1], who synthesized adsorbents using chitosan and used them to remove Pb(II) from aqueous solution, the adsorbent prepared by Gedam with BET surface area of $0.87 \text{ m}^2/\text{g}$ and the pore diameter of 9.77 nm . Small surface area and almost low porosity are characteristics of natural polymer materials, such as cellulose and chitosan, but they have been used as adsorbents in the literature because of their low cost and wide availability, and the results show that their performance in metal ions removal is satisfactory [2,3].

1. Gedam AH, Dongre RS. 2015 Adsorption characterization of Pb (II) ions onto iodate doped chitosan composite: equilibrium and kinetic studies. RSC Adv 5, 54188-54201.

(doi:10.1039/C5RA09899H)

2. Fan L, Chen H, Hao Z, Tan Z. 2012 Cellulose-based macroinitiator for crosslinked poly(butyl methacrylate-co-pentaerythritol triacrylate) oil-absorbing materials by SET-LRP. J Polym Sci Pol Chem 51, 457-462. (doi:10.1002/pola.26404)

3. Rahman ML, Rohani N, Mustapa N, Yusoff MM. 2014 Synthesis of polyamidoxime chelating ligand from polymer-grafted corn-cob cellulose for metal extraction. J Appl Polym Sci 131. (doi:10.1002/app.40833)

It is impossible to draw conclusion about adsorption mechanisms based merely in the trends followed by the data of the adsorption isotherm, particularly when both have a high R^2 value. There are not single evidence of such adsorption mechanisms.

Answer: We have greatly benefited from your constructive comment, considering your comment, we have read many literatures to learned the adsorption isotherm deeply. After that, we revised our manuscript particularly the part of "adsorption isotherm" and marked the changes in red.

Page 6, line 59: "good absorptivity" how come? I think the authors should put attention in the difference of adsorption vs absorption.

Answer: I should apologize for confusing "adsorption" with "absorption", I have corrected this error and marked it in red in the article.

I do not follow the word "sabotage" in line 60 of page 6.

Answer: We feel very sorry about the inaccurate expression. In fact, in line 60 of page 6, what we want to say was that the adsorption capacity was reduced gradually was due to the loosing of some functional groups of the adsorbent by the acid cleavage. We have already improved our expression and we want it to be more accurate and appropriate.

Reviewer: 2

Comments to the Author(s)

The authors made most of the necessary arrangements considering most of the warnings but still some points should be clarified.

1) The circumstances of adsorption experiments should be mentioned e.g. did authors use thermostated water bath shaker or magnetic stirrer? At what rpm? At what temperature?

Answer: Thank you for your constructive comment. Considering your suggestion, we have added the details of the adsorption experiment in the paper, which are described as follows:

"In order to examine the adsorption performance for Pb(II) and Cu(II) of the adsorbent, batch adsorption experiments were carried out on a model BETS-M1 shaker (Kylin-Bell Lab Instruments Co., Ltd., China) with a shaking speed of 120 rpm. In a generally procedure, 10 mg of adsorbent was added to 20 mL of 2.0×10^{-3} mol/L metal ions solution with constant shaking for 24 h at 303.15 K. The effect of pH value on adsorption was studied as pH ranged from 2 to 7. The effect of temperature on adsorption was researched at 293.15 K, 298.15 K, 303.15 K, 308.15 K, 313.15 K, 318.15 K, respectively. The adsorption kinetics were carried out with the initial concentration of 414.4 mg/L for Pb(II) and 128 mg/L for Cu(II) at 303.15 K. Adsorption isotherm were conducted with initial concentrations ranging from 20 to 200 mg/L for Pb(II) and initial concentrations ranging from 15 to 130 mg/L for Cu(II). And then the concentration of Pb(II) and Cu(II) was detected by AAS."

2) The removal efficiency % for both metals should be calculated and added to the abstract and results.

Answer: We have greatly benefited from your suggestion, as you mentioned, it is necessary to calculate the removal efficiency% for metal ions, we have completed the calculations and added it to the abstract and results.

3) *Batch Adsorption Studies: How much metal is lost by adsorption to the walls of the glass flask or the magnetic stirrer? A control without adsorbent should be used.*

Answer: We quite agree with your point about the loss of metal ions by adsorption to the walls of the glass flask or the magnetic stirrer. In fact, we have considered this problem and conducted the controlled experiments without adsorbent, and have eliminated the error in the analysis of the adsorption experiment data based on the results obtained from the control experiment without adsorbent. This is our negligence of the lack of illustration of control experiment without adsorbent in original manuscript, we have added the description of the control experiment to the revised manuscript.

4) *What are the initial concentrations of both metals used in this study?*

Answer: We have to extend sincere apologies for the absence of the initial concentrations. The initial concentrations of Pb(II) and Cu(II) have already supplemented in the adsorption experiments section in the manuscript.

5) *There is no true control used in this work. It would be useful to see the data from the un-treated Corn stalk as a comparison.*

Answer: We are indebted to your kind suggestion. In order to reveal the adsorption performance of modified corn stalk, the adsorption capacity of CS, CS-Br and CS-g-PAN on Pb(II) and Cu(II) were detected to compare with AO CS-g-PAN, and the results were shown in Figure 4 in the revised manuscript.

6) *A very common error in sorption studies is the absence of replicates. How many times were the experiments performed?*

Answer: Thank you for your valuable suggestions. As you say, repeated experiments are important in adsorption performance testing. Indeed, in our adsorption performance experiment, all samples were performed three times and took an average. We apologize for ignoring the description of repeated tests and we have added the detail to the revised manuscript.

7) *It would be more useful to present a table that demonstrates some key parameters (e.g. maximum adsorption capacity) from the previous literature to help make a case that your work is necessary.*

Answer: Thank you for your helpful advice. In order to make a case that our work is necessary, Table 5 was presented to reveal the adsorption performance of AO CS-g-PAN, and compared with other adsorbent which using natural polymers as raw materials and modified by different methods.

8) *Although the authors studied the temperature effect but the thermodynamic parameters were not given. All thermodynamic parameters should be calculated, tabulated and discussed in details.*

Answer: We greatly appreciate your valuable point and we really agree with you, the thermodynamic parameters deserve to be calculated and discussed in detail, which is what we will report in our next article, including the effect of graft rate and graft chain length on adsorption properties, thank you for your valuable suggestion again.

9) *The arrangement of section 3 (results and discussion) is not suitable. All characterization measurements should be reported first and then followed by adsorption results i.e. Surface area and pore size distribution, XPS spectrum analysis and EDS spectrum analysis should be reported directly after Elemental analysis.*

Answer: Thank you for your constructive advice about the arrangement of section 3 (results and discussion), we have made adjustments according to your suggestion in revised manuscript.

Reviewer: 3

Comments to the Author(s)

(1) I do not agree the description "green adsorbent" or "eco-friendly", because almost all the raw materials used are chemicals and even petroleum-based monomers, expect CS.

Answer: We are grateful to your useful suggestions. We have revised our manuscript carefully and corrected the incorrect description such as "green adsorbent" or "eco-friendly", and we hope that the corrections are appropriate.

(2) English needs to be improved. too many errors can be found, such as "pH drift mothed"; is "pore structures" "pore structure parameters"? How do you determine "pore structure"?

Answer: Thank you for your constructive comment, we have found a native speaker to check and rephrase, the grammatical mistakes and typographical errors have been removed, and the errors such as "pH drift mothed" have been corrected in the manuscript; What we want to express is indeed the pore structure parameter, we have to say sorry for these errors, thank you for your kind advice again.

(3) The introduction on Corn stalk is not enough. More information about it should be provided, such as the main chemical composition, ingredients, and so on.

Answer: We benefited a lot from your kind advice, more information about corn stalk including composition and structure has been added to the revised manuscript.

(4) DTG and DSC curves are required to obtain Tmax and reveal the maximum thermal decomposition position.

Answer: We are indebted to you for offering this suggestion. We accepted your comment and the DTG curves were added to the revised manuscript.

(5) The effect of graft ration on adsorption capacity needs to be studied.

Answer: We agree with your point, graft ration is actually an important factor of adsorption capacity, we will investigate the effects of graft rate and graft chain length on adsorption performance in detail in our next paper, thank you for your valuable suggestion again.

(6) A comparison of the adsorbent with other reported adsorbents needs to be made.

Answer: Thank you very much for your kind suggestion. Considering your advice, we make a comparison of the adsorbent with previous literature and the data were shown in Table 5 in the revised manuscript.

Reviewer: 4

Comments to the Author(s)

This study provides an alternative way for adsorbing Pb(II) and Cu(II) by Corn stalk as starting material. The adsorbent exhibited a predominant adsorption performance on Pb(II) and Cu(II). However, the following concerns should be further considered and addressed.

1. The material used here: as “eco-friendly” or “Green”, for instance, line 1 and line 4 in summary. Please check all over the manuscript. If you insist your own opinion, you should compare your material with others to prove your material is “eco-friendly” or “Green”.

Answer: We greatly appreciate the valuable suggestions and we sincerely apologize for the incorrect description such as "green" or "eco-friendly", in fact, almost all the raw materials used are chemicals and even petroleum-based monomers, except CS. We have corrected our manuscript carefully and removed the errors such as “eco-friendly” or “Green”.

2. The manuscript didn't show CS, CS-Br and CS-g-PAN on the adsorption capacity of Pb(II) and Cu(II) to compare their capacity with AO CS-g-PAN.

Answer: We are grateful to you for the kind suggestion. In order to reveal the adsorption performance of modified corn stalk, the adsorption capacity of CS, CS-Br and CS-g-PAN on Pb(II) and Cu(II) were detected to compare with AO CS-g-PAN, and the results were shown in Figure 4 in the revised manuscript.

3. The introduction about adsorption mechanism is not enough, and the authors are suggested to cite this papers:

Junqin Liu, Pingxiao Wu, ShanShan Yang, Saeed Rehman, ZubairAhmed, Nengwu Zhu, Zhi Dang, Zehua Liu. A photo-switch for peroxydisulfate non-radical/radical activation over layered CuFe oxide: Rational degradation pathway choice for pollutants. Applied Catalysis B: Environmental, 2020, 261: 118232.

- Junqin Liu, Pingxiao Wu, Shuaishuai Li, Meiqing Chen, Wentin Cai, Dinghui Zou, Nengwu Zhu, Zhi Dang. Synergistic deep removal of As(III) and Cd(II) by a calcined multifunctional MgZnFe-CO₃ layered double hydroxide: Photooxidation, precipitation and adsorption. *Chemosphere*, 2019, 225: 115-125.
- Shanshan Yang, Zhiyan Huang, Pingxiao Wu, Yihao Li, Xiongbo Dong, Chunquan Li, Ningyuan Zhu, Xiaodi Duan, Dionysios D. Dionysiou. Rapid removal of tetrabromobisphenol A by α -Fe₂O₃-x@Graphene@Montmorillonite catalyst with oxygen vacancies through peroxymonosulfate activation: Role of halogen and α -hydroxyalkyl radicals. *Applied Catalysis B: Environmental*, 2020, 260: 118129.
- Meiqing Chen, Pingxiao Wu, Langfeng Yu, Shuai Liu, Bo Ruan, Haihui Hu, Nengwu Zhu, Zhang Lin. FeOOH-loaded MnO₂ nano-composite: An efficient emergency material for thallium pollution incident. *Journal of Environmental Management*, 2017, 192: 31-38.
- Shanshan Yang, Pingxiao Wu, Junqin Liu, Meiqing Chen, Zubair Ahmed, Nengwu Zhu. Efficient removal of bisphenol A by superoxide radical and singlet oxygen generated from peroxymonosulfate activated with Fe⁰-montmorillonite. *Chemical Engineering Journal*, 2018, 350: 484-495.
- Liya Chen, Pingxiao Wu, Meiqing Chen, Xiaolin Lai, Zubair Ahmed, Nengwu Zhu, Zhi Dang, Yingzhi Bi, Tongyun Liu. Preparation and characterization of the eco-friendly chitosan/vermiculite biocomposite with excellent removal capacity for cadmium and lead. *Applied Clay Science*, 2018, 159: 74-82.
- Shuai Liu, Pingxiao Wu, Meiqing Chen, Langfeng Yu, Chunxi Kang, Nengwu Zhu, Zhi Dang. Amphoteric modified vermiculites as adsorbents for enhancing removal of organic pollutants: Bisphenol A and Tetrabromobisphenol A. *Environmental Pollution*, 2017, 228: 277-286.
- Chongmin Liu, Pingxiao Wu, Yajie Zhu, Lytuong Tran. Simultaneous adsorption of Cd²⁺ and BPA on amphoteric surfactant activated montmorillonite. *Chemosphere*, 2016, 144: 1026-1032.

Answer: Sincerely thank you for offering these papers, we have read these articles carefully and have benefited a lot, the following papers were cited with required form in revised manuscript, they are really helpful.

1. Liu JQ, Wu PX, Yang SS, Rehman S, Ahmed Z, Zhu NW, Dang Z, Liu ZH. 2020 A photo-switch for peroxydisulfate non-radical/radical activation over layered CuFe oxide: Rational degradation pathway choice for pollutants. *Appl. Catal. B: Environ.* 261, 118232. (doi:10.1016/j.apcatb.2019.118232)
2. Chen LY, Wu PX, Chen MQ, Lai XL, Ahmed Z, Zhu NW, Dang Z, Bi YZ, Liu TY. 2018 Preparation and characterization of the eco-friendly chitosan/vermiculite biocomposite with excellent removal capacity for cadmium and lead. *Appl Clay Sci* 159, 74-82. (doi:10.1016/j.clay.2017.12.050)
3. Liu JQ, Wu PX, Li SS, Chen MQ, Cai WT, Zou DH, Zhu NW, Dang Z. 2019 Synergistic deep removal of As(III) and Cd(II) by a calcined multifunctional MgZnFe-CO₃ layered double hydroxide: Photooxidation, precipitation and adsorption. *Chemosphere* 225: 115-125. (doi:10.1016/j.chemosphere.2019.03.009)
4. Liu CM, Wu PX, Zhu YJ, Tran L. 2016 Simultaneous adsorption of Cd²⁺ and BPA on amphoteric surfactant activated montmorillonite. *Chemosphere* 144, 1026-1032. (doi:10.1016/j.chemosphere.2015.09.063)
5. Chen MQ, Wu PX, Yu LF, Liu S, Ruan B, Hu HH, Zhu NW, Lin Z. 2017 FeOOH-loaded MnO₂ nano-composite: An efficient emergency material for thallium pollution incident. *J Environ Manage* 192, 31-38. (doi:10.1016/j.jenvman.2017.01.038)
6. Liu S, Wu PX, Chen MQ, Yu LF, Kang CX, Zhu NW, Dang Z. 2017 Amphoteric modified vermiculites as adsorbents for enhancing removal of organic pollutants: Bisphenol A and Tetrabromobisphenol A. *Environ Pollut*, 228, 277-286. (doi:10.1016/j.envpol.2017.03.082)
7. 62. Yang SS, Wu PX, Liu JQ, Chen MQ, Ahmed Z, Zhu NW. 2018 Efficient removal of bisphenol A by superoxide radical and singlet oxygen generated from peroxymonosulfate activated with Fe⁰-montmorillonite. *Chem Eng J* 350, 484-495. (doi:10.1016/j.cej.2018.04.175)

Reviewer: 5

Comments to the Author(s)

1. I agree with Reviewer 1 that the materials prepared in this study are not "green" or "eco-friendly". The authors made correction but there are still some left in the manuscript:

"eco-friendly" in Summary and Line 59, P9 and "green adsorbent" in Summary.

Answer: We apologize for our negligence. We have corrected our paper, the errors such as “eco-friendly” or “Green” have been removed in revised manuscript.

2. There are some grammar errors and typos in the manuscript. Please find a native speaker to check and rephrase if possible.

For an example, "including the" repeated twice in Line 7, P 11.

And what is "SET-LRP"? The full name of this abbreviation never appeared.

Answer: We are indebted to you for pointing out the suggestion, the manuscript was cross-checked by a native speaker, and his suggestions are also considered for the revised version of the manuscript. We have removed the redundant duplicates such as "including the", and we realized that we should be more careful and serious in our future work. We have to say sorry that there is no full name of the "SET-LRP" in our paper, we have added the full name "single-electron transfer living radical polymerization" in the summary.

3. Materials and Methods

XPS is used to detect the surface composition. For a newly synthesized organic/polymeric material, elemental analysis is usually necessary.

What are the initial concentrations of the adsorbates in batch adsorption tests?

Answer: Thank you very much for pointing out the valuable suggestion, considering your suggestion, we have measured the O, Br and N content of samples using a Vario EL cube elemental analyzer, the result of the elemental analysis and the total exchange capacity (TEC) were illustrated in Table 1 in revised manuscript.

We have supplemented the details of the batch adsorption tests in the revised manuscript, including metal ion concentration, temperature, pH and so on.

4. Results and Discussion

The BET surface area of the obtained material was $< 1 \text{ m}^2/\text{g}$ and the total pore volume around $0.1 \text{ cm}^3/\text{g}$, which implied the small surface and almost nonporous nature of the material. In this case it is questionable whether this material is an appropriate candidate for adsorbent.

Answer: Thank you for your kind comment. Based on the BET surface area of the obtained material, it does have a small surface and almost nonporous nature. But we want to make it clear that the adsorbent we prepared mainly chemical adsorption removal of metal ions, rather than physical adsorption, it can be proved by the adsorption mechanism, namely the formation of new chemical bonds between metal ions and adsorbents. Furthermore, the result was consistent with previous research reported by Gedam[1], who synthesized adsorbents using chitosan and used them to remove Pb(II) from aqueous solution, the adsorbent prepared by Gedam with BET surface area of $0.87 \text{ m}^2/\text{g}$ and the pore diameter of 9.77 nm . Small surface area and almost low porosity are characteristics of natural polymer materials, such as cellulose and chitosan, but they have been used as adsorbents in the literature because of their low cost and wide availability, and the results show that their performance in metal ions removal is satisfactory [2,3].

1. Gedam AH, Dongre RS. 2015 Adsorption characterization of Pb (II) ions onto iodate doped chitosan composite: equilibrium and kinetic studies. RSC Adv 5, 54188-54201.

(doi:10.1039/C5RA09899H)

2. Fan L, Chen H, Hao Z, Tan Z. 2012 Cellulose-based macroinitiator for crosslinked poly(butyl methacrylate-co-pentaerythritol triacrylate) oil-absorbing materials by SET-LRP. J Polym Sci Pol Chem 51, 457-462. (doi:10.1002/pola.26404)

3. Rahman ML, Rohani N, Mustapa N, Yusoff MM. 2014 Synthesis of polyamidoxime chelating ligand from polymer-grafted corn-cob cellulose for metal extraction. J Appl Polym Sci 131. (doi:10.1002/app.40833)

“Thermal degradation behaviors analysis” is not supported by relevant literature. Give the references please.

Answer: We have cited the following relevant literatures to support the “Thermal degradation behaviors analysis”, sincerely thanks again.

1. Chen LY, Wu PX, Chen MQ, Lai XL, Ahmed Z, Zhu NW, Dang Z, Bi YZ, Liu TY. 2018 Preparation and characterization of the eco-friendly chitosan/vermiculite biocomposite with excellent removal capacity for cadmium and lead. Appl Clay Sci 159, 74-82. (doi:10.1016/j.clay.2017.12.050)
2. Yang J, Miranda R, Roy C. 2001 Using the dtg curve fitting method to determine the apparent kinetic parameters of thermal decomposition of polymers. Polym Degrad Stabil 73, 455-461.(doi:10.1016/S0141-3910(01)00129-X)
3. Mohanty S, Verma SK, Nayak SK. 2006 Dynamic mechanical and thermal properties of MAPE treated jute/HDPE composites. Compos Sci Technol 66, 538-547. (doi:10.1016/j.compscitech.2005.06.014)

The adsorption isotherm (q_e vs C_e) in Figure 11 looked more like a linear partition of solute on the material rather than adsorption. And the units are absent in Figure 11a. How did you determine the number of significant figure in Table 3 and 4? Is 5 correct and reasonable?

Answer: Thank you for your kind suggestion. We have seriously considered your comment and consulted some literature, from which we know that both Langmuir and Freundlich isotherms may adequately describe the same set of data for liquid-solid adsorption over a range of concentrations, especially if the solution concentration is small and the adsorption capacity of the solid is large enough to make both isotherm equations approach a linear form. Based on this fact, and in combination with the results

of our adsorption experiments, it is reasonable to determine that the adsorption isotherm (q_e vs C_e) is adsorption rather than a linear partition of solute on the material.

The units in Figure 11a (Figure 12a in the revised manuscript) have been added in revised manuscript.

We have read a mass of article to determine the number of significant figure, in these article, the number of significant figure of pseudo-first order kinetics model and pseudo-second order kinetics model is 4, and the number of significant figure of Langmuir isotherm and Freundlich isotherm is 5, Based on literatures and in the principle of accuracy and rationality, we determine the number of significant figure in Table 3 and 4.

What are the equilibrium concentrations of the adsorbate corresponding to the equilibrium (maximum) adsorption capacity 231.84 mg/g and 94.72 mg/g respectively? It won't make sense to compare the adsorption capacity without mentioning the equilibrium concentration.

Answer: We greatly appreciate your valuable suggestion, the equilibrium concentrations of the adsorbate corresponding to the equilibrium (maximum) adsorption capacity 231.84 mg/g and 94.72 mg/g were 298.48 mg/L and 79.64 mg/L, in order to reveal the adsorption capacity clearly, we calculated the metal ion removal efficiency by equilibrium concentration, as shown in figure 4 in the revised manuscript.

The "Effect of the dosage of adsorbent" part is unnecessary and superfluous now that the adsorption isotherms are obtained.

Answer: We are indebted to your kind suggestion, we accept your advice and removed the "Effect of the dosage of adsorbent", sincerely appreciate your kind comment.